# Oligoclonal CD4+CXCR5+ T cells with a cytotoxic phenotype appear in tonsils and blood
Chunguang Liang [1,10], Silvia Spoerl [2], Yin Xiao[3], Katharina M. Habenicht [4], Sigrun S. Haeusl [3], Isabel Sandner[2], Julia Winkler[2], Nicholas Strieder [5], Rüdiger Eder[6], Hanna Stanewsky[5], Christoph Alexiou[7], Diana Dudziak [8,10], Andreas Rosenwald[3,9], Matthias Edinger[5,6], Michael Rehli [5,6], Petra Hoffmann [5,6,11], Thomas H. Winkler [4,11] & Friederike Berberich-Siebelt [3,11] ✉

In clinical situations, peripheral blood accessible CD3+CD4+CXCR5+ T-follicular helper ($T_{FH}$) cells may have to serve as a surrogate indicator for dysregulated germinal center responses in tissues. To determine the heterogeneity of $T_{FH}$ cells in peripheral blood versus tonsils, CD3+CD4+CD45RA−CXCR5+ cells of both origins were sorted. Transcriptomes, TCR repertoires and cell-surface protein expression were analysed by single-cell RNA sequencing, flow cytometry and immunohistochemistry. Reassuringly, all blood-circulating CD3+CD4+CXCR5+ T-cell subpopulations also appear in tonsils, there with some supplementary $T_{FH}$ characteristics, while peripheral blood-derived $T_{FH}$ cells display markers of proliferation and migration. Three further subsets of $T_{FH}$ cells, however, with bona fide T-follicular gene expression patterns, are exclusively found in tonsils. One additional, distinct and oligoclonal CD4+CXCR5+ subpopulation presents pronounced cytotoxic properties. Those 'killer $T_{FH}$ ($T_{FK}$) cells' can be discovered in peripheral blood as well as among tonsillar cells but are located predominantly outside of germinal centers. They appear terminally differentiated and can be distinguished from all other $T_{FH}$ subsets by expression of NKG7 (TIA-1), granzymes, perforin, CCL5, CCR5, EOMES, CRTAM and CX3CR1. All in all, this study provides data for detailed CD4+CXCR5+ T-cell assessment of clinically available blood samples and extrapolation possibilities to their tonsil counterparts.

Follicular helper T ($T_{FH}$) cells are a subgroup of highly specialized CD4+CXCR5+ (C-X-C chemokine receptor 5) T cells, which drive germinal center (GC) formation and responses[1,2]. In GCs, $T_{FH}$ provide cognate help to GC-B cells, which compete for $T_{FH}$ help through increased affinity for antigen and subsequent presentation. Expression of CXCR5 – and downregulation of CCR7 and PSGL1 – is essential for pre-$T_{FH}$ cells to get into contact with B cells at the T-cell/B-cell border of follicles and to form a GC[3,4]. Both CXCR5+ B and CXCR5+ T cells follow a gradient of the chemokine CXCL13, which is the selective chemoattractant produced by follicular dendritic cells and—in humans additionally – by $T_{FH}$ cells themselves[1,5]. Besides CXCL13, human $T_{FH}$ cells secrete limited amounts of IL-21 and IL-4 to provide help to GC-B cells in synaptic contact. Further cell-cell contact is enabled by CD28, ICOS, CD40L, PD-1, OX-40 and SLAM family receptors. Commonly, CD4+CXCR5+PD-1+ ones are regarded as pre-$T_{FH}$ cells still residing in the mantle zone and CD4+CXCR5hiPD-1hi as true

GC-$T_{FH}$ cells. High PD-1 expression inhibits T-cell recruitment into the follicle, which is directly opposed by ICOS signaling[6]. Besides, PD-1 diminishes GC-$T_{FH}$-cell differentiation and expansion allowing competition between GC-B cells for help. The latter is further ensured by PD-1-mediated blocking of TCR signaling and thereby heightening the stringency of GC-affinity selection. On the other hand, PD-1 promotes IL-21 expression and follicle → GC homing[6]. Like GC-B cells, all CXCR5+ $T_{FH}$ cells express BCL-6 as their key transcription factor[7,8].

Circulating (c) CXCR5+ $T_{FH}$ cells downregulate BCL-6[9]. Being CCR7+CD62L+, they are $T_{FH}$ cells in the central memory state thought to be programmed for preferential recruitment to follicles to induce plasma cell differentiation. Accordingly, they readily secrete CXCL13, IL-21 and IL-10 upon activation[10]. Early microarray analyses revealed that tonsillar CXCR5+ICOS+ $T_{FH}$ cells are highly distinct from CXCR5+ICOS+ c$T_{FH}$ cells[11]. Three subclasses of human c$T_{FH}$ cells have been defined based on

their differential co-expression of chemokine receptors[12]. $CXCR5^+CXCR3^+$ $T_{FH}1$ secrete IFN-γ and have only limited isotype-switching capabilities, whereas IL-4-secreting $CXCR5^+CXCR3^-CCR6^-$ $T_{FH}2$ and IL-17-secreting $CXCR5^+CCR6^+$ $T_{FH}17$ cells can mediate class switching in vitro[12,13]. Differential expression of ICOS, PD-1, and CCR7 further defines functionally distinct $cT_{FH}$ subpopulations, with activated $CCR7^{lo}ICOS^+PD-1^{hi}$ or $CCR7^{hi}CCR6^{hi}PD-1^{hi}$ superior in supporting in vitro antibody production[10,13]. Finally, bulk sequencing revealed that only $CD4^+CXCR5^+CXCR3^-PD-1^+$ PBMCs express some $T_{FH}$ signature genes such as *MAF, POU2AF1* (encoding BOB1/OBF1), *TIGIT*, and *SLAMF6*, but not *BCL6*[14]. Of note, also $CD4^+PD-1^+$, but $CXCR5^-BCL-6^{low}$ T-peripheral helper cells ($T_{PH}$), detected in the blood and inflamed tissues of several autoimmune-diseased patients, exert some B-helper functions[15].

The relationship between $cT_{FH}$ and $GC-T_{FH}$ cells remains enigmatic, although genetic studies suggest that $CXCR5^+CD4^+$ $cT_{FH}$ cells are predominantly generated from cells committed to the $T_{FH}$ lineage, but not from $GC-T_{FH}$ cells[16,17]. Nevertheless, $GC-T_{FH}$ cells can acquire a memory state within secondary lymphoid organs (SLO) and then downregulate BCL-6, CXCR5, and PD-1 while upregulating CD127, CCR7, and CD62L[18].

GC-B and $T_{FH}$ cells are controlled by T-follicular regulatory ($T_{FR}$) cells, which are $FOXP3^+$ T-regulatory cells (Tregs) acquiring characteristics of $T_{FH}$ cells including CXCR5 upregulation[19–24], or may derive from conventional T, including $T_{FH}$ ($iT_{FR}$), cells[25,26]. Dysregulation of either $T_{FH}$ or $T_{FR}$ cells can cause loss of immune tolerance with abnormal, high levels of autoantibodies, which has been implicated in the pathogenesis of several autoimmune diseases[2,17,19]. Since clinical study markers are mostly assessed in the easily accessible, human peripheral blood (PB), it is essential to understand what phenotypes of particular T-cell subsets – including $cT_{FH}$ cells –[17,27] can be expected in comparison to SLOs. Thus, we set out to analyze in detail the $CD3^+CD4^+CXCR5^+$ T-cell subpopulations in tonsils and PB, making use of CXCR5 as the most stable marker for both $GC-T_{FH}$ and $cT_{FH}$ cells[28]. Four subpopulations with variations of the classical $GC-T_{FH}$ phenotype were defined in tonsils, only one of which was also present among $cT_{FH}$ cells. In contrast, all $CD3^+CD4^+CXCR5^+$ T-cell subsets identified in PB were also found in tonsils, there with additional characteristics of $T_{FH}$ cells. Strikingly, a distinct and oligoclonal $CD3^+CD4^+CXCR5^+$ subpopulation exhibited unambiguous cytotoxic properties.

## Results

### All identified $T_{FH}$ and $T_{FH}$-like subtypes are present in tonsils, whereas $cT_{FH}$ cells do not comprise all $CD3^+CD4^+CD45RA^-CXCR5^+$ T-cell clusters

Flow cytometry-sorted live (DAPI$^-$) $CD3^+CD4^+CD45RA^-CXCR5^+$ T cells from PB and tonsils were subjected to scRNAseq (Supplementary Fig. 1a–c). Tonsils were collected from three donors of different ages, either male or female, with no, mild, and recurrent / moderate tonsillitis (Supplementary Table 1). To increase the likelihood of $cT_{FH}$ cells[29], these were isolated from PBMCs not only before but also after routine booster vaccination (tetanus toxoid) of two healthy individuals, one male and one female. We further included $cT_{FH}$ cells from PBMCs of patients undergoing allogeneic hematopoietic stem cell transplantation (allo-HSCT) before and after cyclosporin A (CsA) tapering. As an inhibitor of the phosphatase calcineurin, CsA interferes with TCR-mediated NFAT activation resulting in suppression of donor-derived T cells[30]. After tapering of CsA, $cT_{FH}$ cells can re-activate. Since $T_{FH}$ cells are defined as $CD45RA^-$, CD45RA was part of the sorting to exclude any naïve or terminally differentiated effector memory cells re-expressing CD45RA ($T_{EMRA}$) $CD3^+CD4^+$ T cells.

After dimension reduction and visualization in UMAPs, the result indicated that the canonical correlation analysis (CCA) method was effective in removing the batch-effect between different samples (Supplementary Fig. 2a–d). $CD3^+CD4^+CD45RA^-CXCR5^+$ T cells distributed into 12 clusters (c0-c11) (Fig. 1a). Relative cluster sizes were similar in tonsils or PBMCs but differed considerably in tonsils vs PB (Fig. 1b, Supplementary Fig. 3a–d). C5 and c11 were almost exclusively present in tonsils. Likewise, c10 was

dominated by tonsillar T cells, while c9 – despite a high contribution from tonsillar cells – included cells from all origins. In sum, only tonsillar cells were found in all clusters.

### $CD3^+CD4^+CD45RA^-CXCR5^+$ T cells present with described and unknown $T_{FH}$ transcriptomes

The top differentially expressed transcripts highlighted the clustering (Fig. 1c, Supplementary Data 1). Together with the expression pattern of known T-follicular genes, they revealed the cluster phenotypes (Fig. 1d; Supplementary Fig. 4). C2 with only 0.3% of all 72,608 cells analyzed, contained actively proliferating cells with high expression of *MKI67*, microtubule-associated proteins (*STMN1, TUBB*) and cell cycle progression markers (*PCNA, HMGB1/2*). The RNA expression pattern of c0, which comprised two thirds of all cells, was indicative of central memory (*IL7R*) T cells with an enhanced likelihood of entry into cell cycle (*MYC*), but no indication of active proliferation. *LEF1* expression in addition to *TCF7* pointed towards $T_{FH}$ differentiation[31]. Although dominant with respect to all cells, c0 constituted only one third of $CD4^+CXCR5^+$ T cells in tonsils (Supplementary Fig. 3b). This was in line with high levels of *CCR7 and S1PR1* mRNA in this cluster[32], i.e. progenitor (pro)$T_{FH}$ cells circulating in (*CCR7*) and out (*S1PR1*) of SLOs.

With a pronounced MHCII expression comparable to c2, c1 resembled activated T cells, while abundant *STAT1* and *CXCR3* defined their $T_{FH}1$ phenotype. On the contrary, cells in c3 and c4 exhibited a strong or weak $T_{FH}17$ phenotype, presumably entering the cell cycle (*MYC*) in c3 also expressing *CCR6, KLRB1* (CD161)[33] and *RORA*, while those in c4, besides expressing *RORA*, displayed long non-coding RNA *NEAT1*[34] as well as the migration-enabling and $T_{FH}$ phenotype-preceding *KLF2*[35,36]. C3 – together with c0 – displayed most *IL7R* transcripts, which is upregulated on T-follicular cells upon egress from SLOs or $GC-T_{FH}$ memory formation[18,37].

With *STAT1, STAT2* and *IRF7, MXI, GBP5, ISG15* and *ISG20, IFI6, IFITM1* and *IFITM2*, c8 exhibited a signature of type I interferon response – an unappreciated $T_{FH}$-phenotype – pointing to responses to viral infections[17].

None of the so far mentioned clusters, however, displayed a canonical $T_{FH}$ gene expression pattern[1]. Yet, a refined heatmap (Supplementary Fig. 5) demonstrated more prominent *BCL6* in c5, c9-c11 and heightened *CXCR5* in c5 and c10. Indeed, while c9-c11 expressed elevated *MAF*, c10 additionally showed enhanced *ICOS, PDCD1* (encoding PD-1), *TIGIT*[38], *CXCL13* and *IL21* (Fig. 1d, Supplementary Fig. 4 and Supplementary Data 1). Cells in c5 also expressed robust *PDCD1, TIGIT* and *CXCL13*, but had exchanged the dominance of the transcriptional regulator *MAF* for *BATF*[39], *POU2AF1*[40], *TOX* and *TOX2*[41,42] and added advanced *CD200*[43] to their inhibitory receptor collection. In agreement with our and other data, all clusters with a more classical $T_{FH}$-gene expression profile, i.e., c5, c9-c11, displayed higher levels of CXCR5-supportive *NFATC1* (Fig. 1d)[11,20,44,45]. Similarly, all these clusters showed enhanced *NR4A2* and c11 also *NR4A1*, encoding transcription factors upstream and downstream of BCL-6 expression, yet not essential in $T_{FH}$ cells[46]. Overall, Fig. 1c made clear that $CD3^+CD4^+CXCR5^+$ cells in c9-c11, in contrast to all others, did not express the eukaryotic translation elongation factor 1 (*EEF1A/B*) nor lactate dehydrogenase B (*LDHB*) or the proteolipid Myelin and lymphocyte protein (*MAL*) involved in LCK recruitment to the TCR complex. Along with c5 cells, they also showed low expression of *VIM*, a cytoskeletal component involved in low-density lipoprotein transport, but high expression of *SLC2A3* encoding the glucose transporter GLUT3.

The $CD3^+CD4^+CD45RA^-CXCR5^+$-sorted cells in c6 were expressing *FOXP3, IZKF2* (encoding HELIOS), *BATF* and *TIGIT*, resembling a profile partially shared with effector Treg (*TIGIT*)[47] and tissue-resident Treg (*BATF*)[48] cells.

Altogether, applying scRNAseq to PB- and tonsil-derived $CD3^+CD4^+CD45RA^-CXCR5^+$ T cells, we identified clusters with activation / proliferation / central memory features, with $T_{FH}1$, $T_{FH}17$ and $T_{FH}17$-like cells, one with an IFN type I signature, four clusters with variations of classical $T_{FH}$ cells and one less classical $T_{FR}$ cluster.

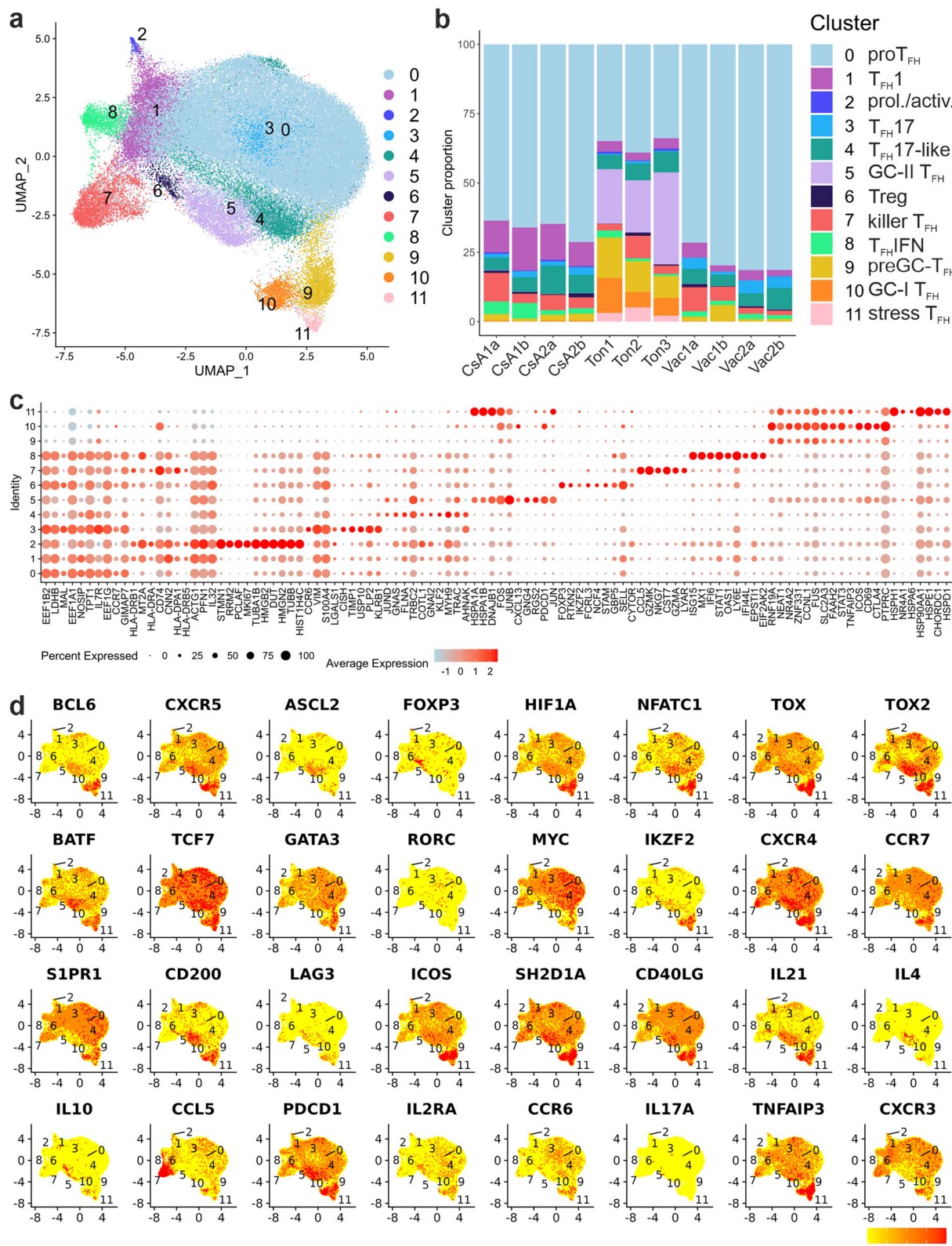

**Fig. 1 | Transcriptional landscape of CD3⁺CD4⁺CXCR5⁺ cells derived from PB and tonsils.** DAPI⁻CD4⁺CD45RA⁻CXCR5⁺ were flow cytometry-sorted prior to scRNAseq; 3 tonsils as well as paired PBMC samples from either before (Vac1a; Vac2a) and after (Vac1b; Vac2b) booster vaccination or from recovered allo-HSCT PBMCs during (CsA1a; CsA2a) and after (CsA1b; CsA2b) CsA prophylaxis. **a** UMAP of pooled CD4⁺CXCR5⁺ T cells ($n$ = 72,608 cells). Each dot corresponds to a single cell, color-indexed according to its cluster affiliation (c0-c11).

**b** Quantification of the cluster distribution; the y-axis of the bar graph depicts the relative proportion of each cluster (c0-c11) within individual samples (cluster annotation according to the individual transcriptomes). **c** Bubble plots to project gene markers of each cluster. The size of a dot corresponds to the percentage of cells expressing the # feature in each cluster. The color represents the average expression level. **d** Feature plots of chosen gene markers on simplified UMAPs.

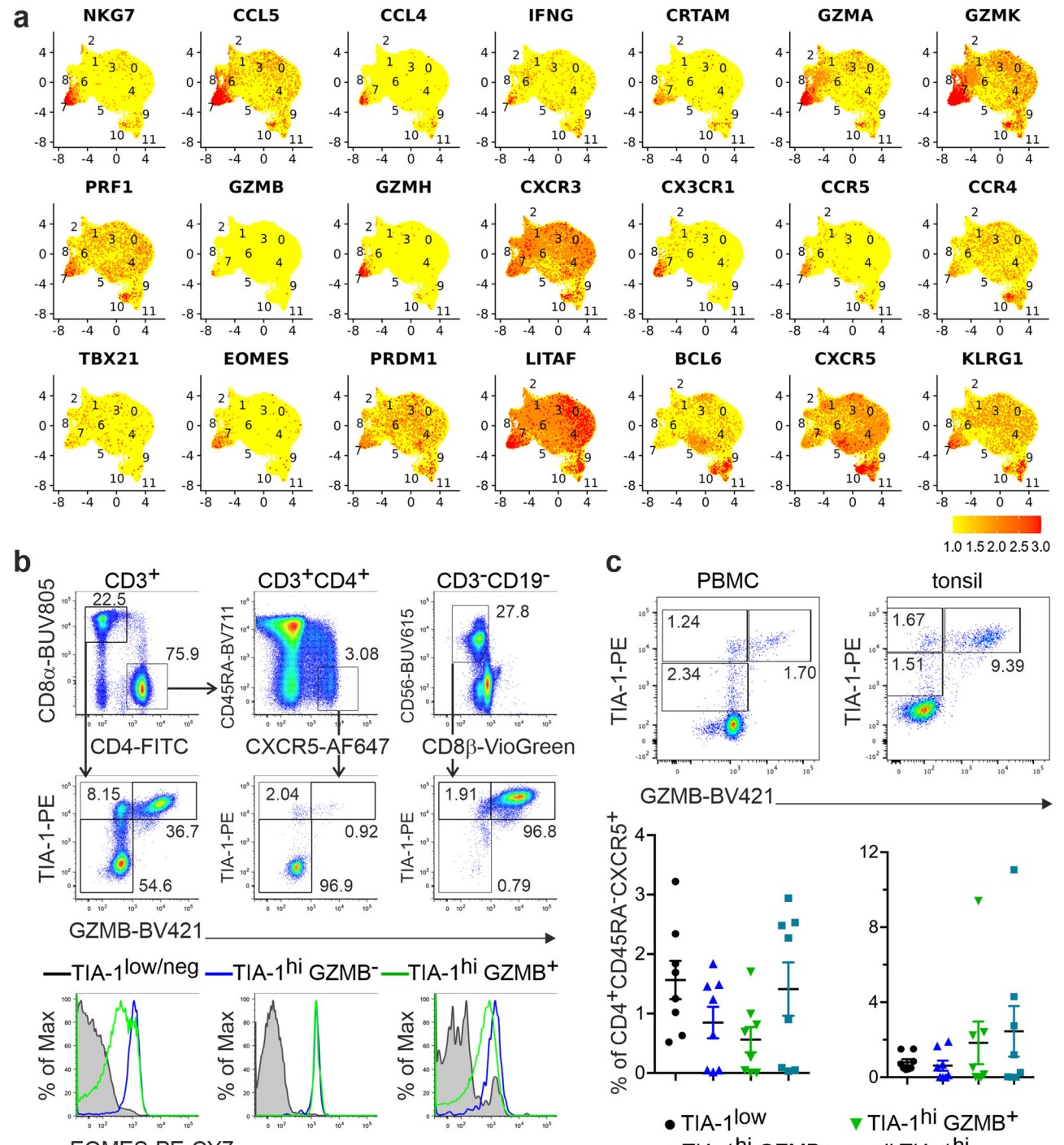

**Fig. 2 | Peripheral and tonsillar $T_{FH}$-like cells exhibit a cytotoxic phenotype.**
**a** Feature plots of chosen gene markers on simplified UMAPs. **b** Cells were isolated from PB by Ficoll, stained immediately and gated for CD3$^+$CD19$^-$ (upper left plot), CD3$^+$CD4$^+$ (upper central plot) or CD3$^-$CD19$^-$ (upper right plot). TIA-1 and GZMB expression levels were assessed in CD8$^+$ T cells (left middle plot), CD4$^+$CXCR5$^+$CD45RA$^-$ $T_{FH}$ cells (central middle plot) and CD56$^+$ NK cells (right middle plot). EOMES expression was determined in respective TIA-1$^{low/neg}$, TIA-1$^{hi}$GzmB$^-$ and TIA-1$^{hi}$GzmB$^+$ subpopulations

of CD8$^+$ T cells (left lower plot), $T_{FH}$ cells (central lower plot) and NK cells (right lower plot). **c** Leukocytes from PB or from tonsils, stained immediately and gated for CD3$^+$CD19$^-$CD4$^+$CD8a$^-$CD45RA$^-$CXCR5$^+$. Proportions of TIA-1$^{low}$GZMB$^-$ and TIA-1$^{hi}$GZMB$^-$ or TIA-1$^{hi}$GZMB$^+$ subpopulations are depicted. Upper plots show examples with pronounced $T_{FK}$ populations, lower scatter plots show individual data points and means ± SEM of $n = 8$ donors each.

## Some peripheral and tonsillar $T_{FH}$-like cells exhibit a cytotoxic phenotype

Additionally, we detected a CD3$^+$CD4$^+$CD45RA$^-$CXCR5$^+$ T-cell cluster (c7) with a manifest cytotoxic signature. C7 was most prominent in the tonsil from the patient with recurrent tonsillitis (Ton2) and consistently present among c$T_{FH}$ cells. Dominant transcripts were *CCL5*, *GZMA* and

*GZMK*, *NKG7* and *CST7* (Figs. 1c, 2a). Perforin and other granzymes were cluster-defining as well. CCL5 shows the highest affinity for CCR5, which is generally well expressed on CD4$^+$ T cells and here particularly high on cells in c7. NKG7 (Natural Killer Cell Granule Protein 7) regulates cytotoxic granule exocytosis[49], while cystatin F (encoded by *CST7*), expressed in NK and CD8$^+$ T cells as well as in other CD4$^+$ cytotoxic T cells, controls perforin

and granzyme maturation[50]. Further support for the CD4+ cytotoxic phenotype was the abundance of both the adhesion molecule *CRTAM* and the transcription factor *EOMES*[51,52]. Preferential expression of *CXCR3* together with *CX3CR1* and *KLRG1* indicated type 1 terminal differentiation. Interestingly, well-expressed *LITAF* or *PRDM1*-encoded transcriptional regulators are in reciprocal negative loops with BCL-6[7,53,54], coinciding with just moderate *CXCR5* and *PDCD1* expression (Figs. 1d and 2a).

We verified co-expression of NKG7 (recognized by a monoclonal antibody against T-cell intracellular antigen 1, TIA-1) with granzyme B in a subpopulation of $cT_{FH}$ cells, which resembled CD8+ T and NK cells in this respect (Fig. 2b). Irrespective of GZMB co-expression, all TIA-1+CD4+CXCR5+ T cells stained positive for EOMES. The appearance of CD4+CXCR5+TIA-1+GZMB+ T cells in several PB and tonsil donors validated their presence in periphery and SLOs, while being more prevalent in tonsils (Fig. 2c; Supplementary Data 2).

In sum, a CD3+CD4+CD45RA−CXCR5+ T-cell cluster, present in both PB and tonsils, displayed a clear cytotoxic gene expression signature. Thus, we termed these cells 'killer $T_{FH}$' ($T_{FK}$) cells.

### $T_{FK}$ cells express CXCR3 and PD-1

To enrich the transcriptome analysis with actual protein data, we determined the surface expression of PD-1, CD25, CXCR3 and CCR6 by CITEseq. CD25 surface expression showed, as expected, a highly positive correlation with *FOXP3* RNA from c6 but also a strong negative correlation with c5, c9-c11 and – of note – c7 (Fig. 3a). PD-1 accumulated in c5 and 9-11, while either CXCR3 or CCR6 enriched in the other clusters. Protein expression of these chemokine receptors defining $T_{FH}1$ and $T_{FH}17$ appeared even lower in c9-c11 than the corresponding RNA expression, which could be due to an auxiliary posttranslational regulation[55]. Pairwise comparison revealed low PD-1 on CD25+ cells and vice versa, a dominance of CXCR3 on $T_{FH}1$ c1, $T_{FH}IFN$ c8 and $T_{FK}$ c7 cells, opposed by a high CCR6 expression on $T_{FH}17$ c3 and $T_{FH}17$-like c4 (Fig. 3b).

To verify the dominant surface expression of CXCR3 *vs* CCR6 on $T_{FK}$ cells, we applied flow cytometry and added anti-CD57 reported to indicate a weak cytotoxic phenotype in CD4+ T cells, including $T_{FH}$ cells[56]. CD4+CD45RA−CXCR5+TIA-1+ $T_{FK}$ cells were predominantly CD57− but exhibited co-expression of CXCR3 and PD-1, while CCR6 was barely detectable, thereby confirming the CITEseq and scRNA data (Fig. 3c, d; Supplementary Fig. 6a, b; Supplementary Data 2). Interestingly, the frequency of TIA-1+PD-1+CXCR3+ cells was significantly higher among CD4+CD45RA−CXCR5+ $T_{FH}$ cells than among CD4+CD45RA+CXCR5+/− or even CD4+CD45RA−CXCR5− T cells.

### TIA-1+ $T_{FH}$ cells degranulate upon SEB activation

To address whether $T_{FK}$ cells exert cytotoxicity in an MHCII-restricted manner, we applied the 'cytokine-independent activation-induced marker' (AIM) method, which identifies Ag-specific GC-$T_{FH}$ cells[57]. We added staphylococcal enterotoxin B (SEB) to whole tonsillar cell cultures and identified $T_{FH}$ cells by CD4, CXCR5 and PD-1 after four days. Indeed, a significant fraction of TIA-1+ $T_{FH}$ cells, without or with still detectable GZMA, showed CD107a on their surface, indicating recent degranulation of their cytotoxic granulae[58] (Fig. 4a; Supplementary Data 2).

### Removal of calcineurin inhibition after allo-HCT reduces the frequency of $cT_{FK}$ cells

To understand the impact from vaccine boosting and CsA tapering, respectively, we performed analyses on samples from the same donors, i.e., Vac2a *vs* Vac2b and CsA1a *vs* CsA1b. Booster vaccination relatively increased c4 $T_{FH}$ 17-like cells at the expense of c1 $T_{FH}1$, c7 $T_{FK}$ and c8 $T_{FH}IFN$ (Fig. 4b). To understand a possible difference in gene expression profiles, we carried out a pseudo-bulkRNAseq analysis using our scRNAseq data. Only minor changes such as stronger *CXCR4* expression in Vac2a or higher *CCND3* (encoding CyclinD3) in Vac2b could be observed (Fig. 4c).

Releasing the T cells from calcineurin inhibition allowed an enhanced proportional size of c1 $T_{FH}1$, c3 $T_{FH}17$ and c8 $T_{FH}IFN$, while that of c7 $T_{FK}$

cells clearly diminished (Fig. 4b). Comparison of gene expression in Seurat showed *CST7*, *GZMA*, *GZMK vs IL7R*, *CCR7*, and *MAL* as the highest ranked transcripts. This agreed with major gene markers of c7 being downregulated, while central memory (*IL7R*, *CCR7*) or even naïve (*MAL*[59]) signatures were expanded (Fig. 4d). This pattern could be observed in c1, c7, and c8 also individually (Fig. 4e), in sum indicating that tapering of CsA resets T cells, and in particular $cT_{FH}$ cells, to a more naïve or $T_{CM}$ differentiation stage lessening a possible dominance of $T_{FK}$ cells.

### All CD3+CD4+CD45RA−CXCR5+ T-cell clusters subdivide depending on their tissue origin

The clusters in Supplementary Fig. 5 sub-clustered. We therefore wondered whether these differences were tissue-specific. Indeed, heatmaps for individual clusters comparing PB-derived *vs* tonsillar cells demonstrated differentially expressed transcripts for either tissue (Supplementary Fig. 7a). Compared to their $cT_{FH}$ counterparts in the same cluster, tonsil-derived CD3+CD4+CD45RA−CXCR5+ T cells predominantly displayed a canonical $T_{FH}$-transcriptome[1,25] as exemplified by transcripts for TOX, TOX2, BATF, CXCR5, PD-1, CD200, ICOS, SAP (*SH2D1A*), FOS, JUNB and NURR1 (*NR4A2*), which were highly significant in most clusters (Fig. 5a, Supplementary Fig. 8). Besides, tonsillar CXCR5+ Treg cells exposed elevated transcript levels for $T_{FR}$-typical IL-1R2, Treg/$T_{FR}$-essential CTLA4 and CD25 (*IL2RA*)[20,21,60,61], or resembled CD25− GC-$T_{FH}$ cells, upregulating FOXP3 for GCR termination[62]. *SELPLG* (encoding PSGL1), *MYC*, *CXCR3* and/or *CCR6* characterized the $cT_{FH}$ equivalents best. Of note, GLUT3 (*SLC2A3*) indicated a tonsil origin (Supplementary Fig. 7b). $T_{FK}$ cells followed the same pattern but were unique in retaining some *CXCR3* in the tonsils (Fig. 5a, b).

### $T_{FK}$ cells are preferentially located outside of germinal centers

To determine the localization of the $T_{FK}$ cells with regard to GCs, we stained tonsil samples to define the GC with dark, light and mantle zone as well as CD19, CD4 and TIA-1 together with BCL-6 or CXCR5 for $T_{FK}$ identification (Fig. 6a, Supplementary Fig. 9). Consistent with *SELPLG* and *CXCR3* expression observed in all cells of c7, we could unambiguously detect single $T_{FK}$ cells localized close to the edge or outside GCs. In a more objective way, we quantitated $T_{FK}$ cells within GCs in comparison to their occurrence at the T-B border (Supplementary Fig. 10). While some images showed no $T_{FK}$ cells at all, they were overall significantly higher at the T-B border than within the follicles (Fig. 6b; Supplementary Data 2).

### $T_{FK}$ cells are oligoclonal

T cells have a natural barcoding feature, the TCR sequence, which allows tracking of their clonal relationship. Analysis of the TCR repertoires revealed that $T_{FK}$ cells (c7) displayed by far the highest degree of clonality defined by the largest clonal size of all T-cell subsets identified (Fig. 7a, Supplementary Fig. 11a). Remarkably, the TCR diversity increased upon booster vaccination and especially after CsA tapering in allo-HSCT recipients, although some clones persisted (Fig. 7b–d, Supplementary Figs. 11b, 11c, and Supplementary Fig. 12). Correspondingly, the dominance of $T_{FK}$ clones was shared by $T_{FH}1$ after CsA tapering and it was overruled by $proT_{FH}$ from c0 after booster vaccination (Fig. 7e). While all tonsils harbored clones of GC-$T_{FH}$ cells, Ton2 was dominated by a few fairly large clones and one dramatically expanded clone with a $T_{FK}$ transcriptome. Thus, $T_{FK}$ cells showed a surprisingly high level of clonality in both SLOs and PB, which was unique to this subpopulation.

### Both classical $T_{FH}$ and $T_{FK}$ cells share TCR clones with $proT_{FH}$ cells and with each other

RNA velocity allows determination of the developmental relationship of clusters and was examined for Ton2 and CsA1a, i.e., for samples with the most prominent $T_{FK}$ population in either tonsil or PB. The streamlines indicated that c0 is the progenitor of most $T_{FH}$ cells likely to develop into $T_{FH}1$, $T_{FH}17$, $T_{FH}17$-like and $preGC-T_{FH}$ of c9 (Fig. 8a). C10/GC-I $T_{FH}$ give rise to c11/stressed $T_{FH}$ cells, whereas c10/GC-I itself and c5/GC-II $T_{FH}$ cells

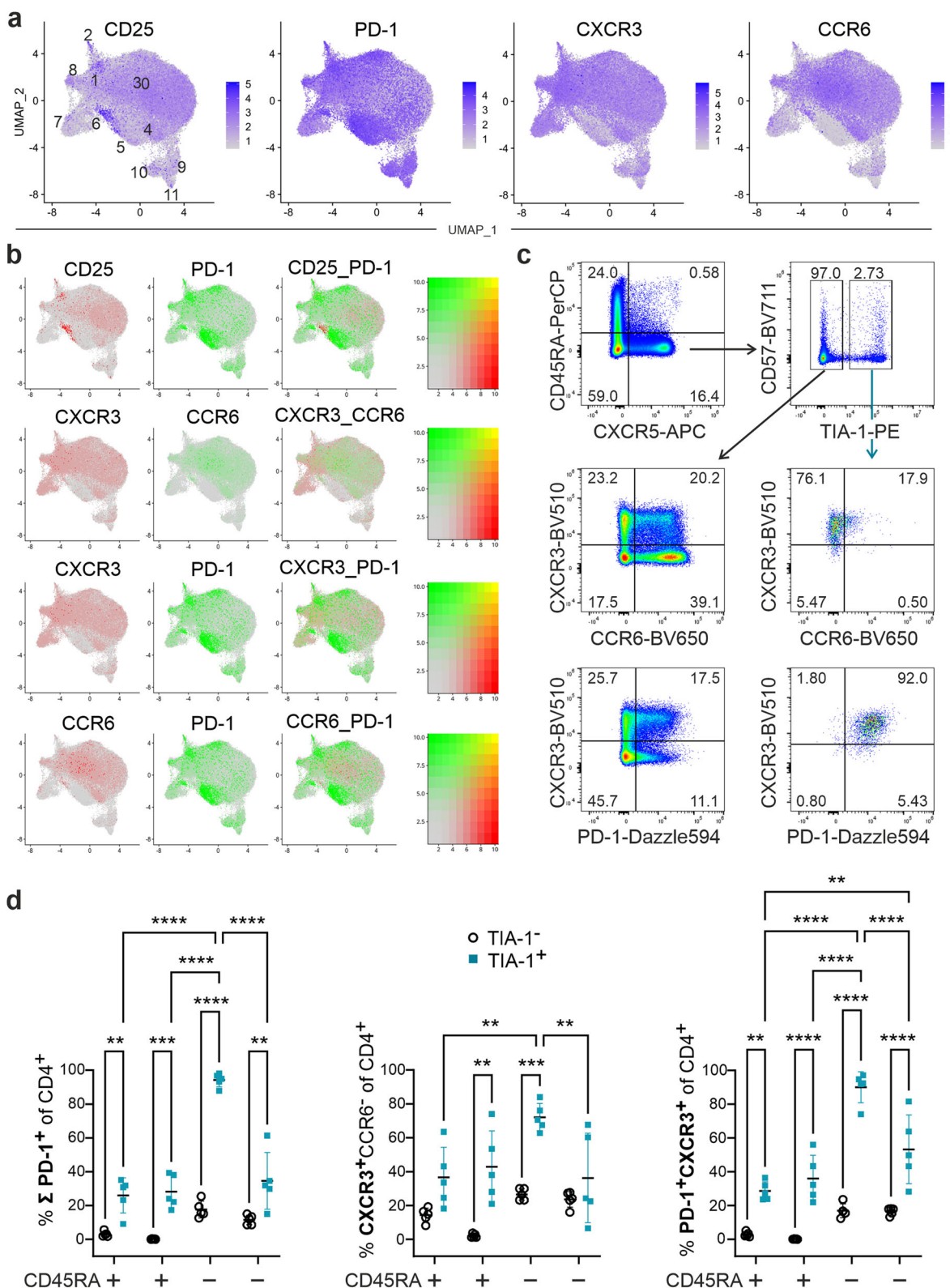

**Fig. 3 | CD4⁺CXCR5⁺ T cells from *c7* express PD-1 and CXCR3, but hardly CD25 or CCR6. a** UMAP visualization of samples with 4 CITEseq antibody-derived tags (ADT). **b** Feature plots for a pairwise comparison of ADT distribution. **c, d** Leukocytes isolated from PB by Ficoll, stained immediately and gated for CD3⁺CD4⁺ and CD45RA⁻ or ⁺ as well as CXCR5⁻ or ⁺. The frequencies of CXCR3 *vs* CCR6 and CXCR3 *vs* PD-1 of TIA-1⁻ and TIA-1⁺ cells are depicted in representative dot plots for CD3⁺CD4⁺CD45RA⁻CXCR5⁺ T_FH cells (**c**). ∑PD-1⁺, CXCR3⁺CCR6⁻, and PD-1⁺CXCR3⁺ double expression are shown as cumulative graphs for TIA-1⁻ and TIA-1⁺ CD4⁺ CD45RA⁺/⁻ CXCR5⁺/⁻ cells (two-way ANOVA with Šídák post-hoc analysis, *n* = 5 donors; **p ≤ 0.01, ***p ≤ 0.001, ****p ≤ 0.0001) (**d**).

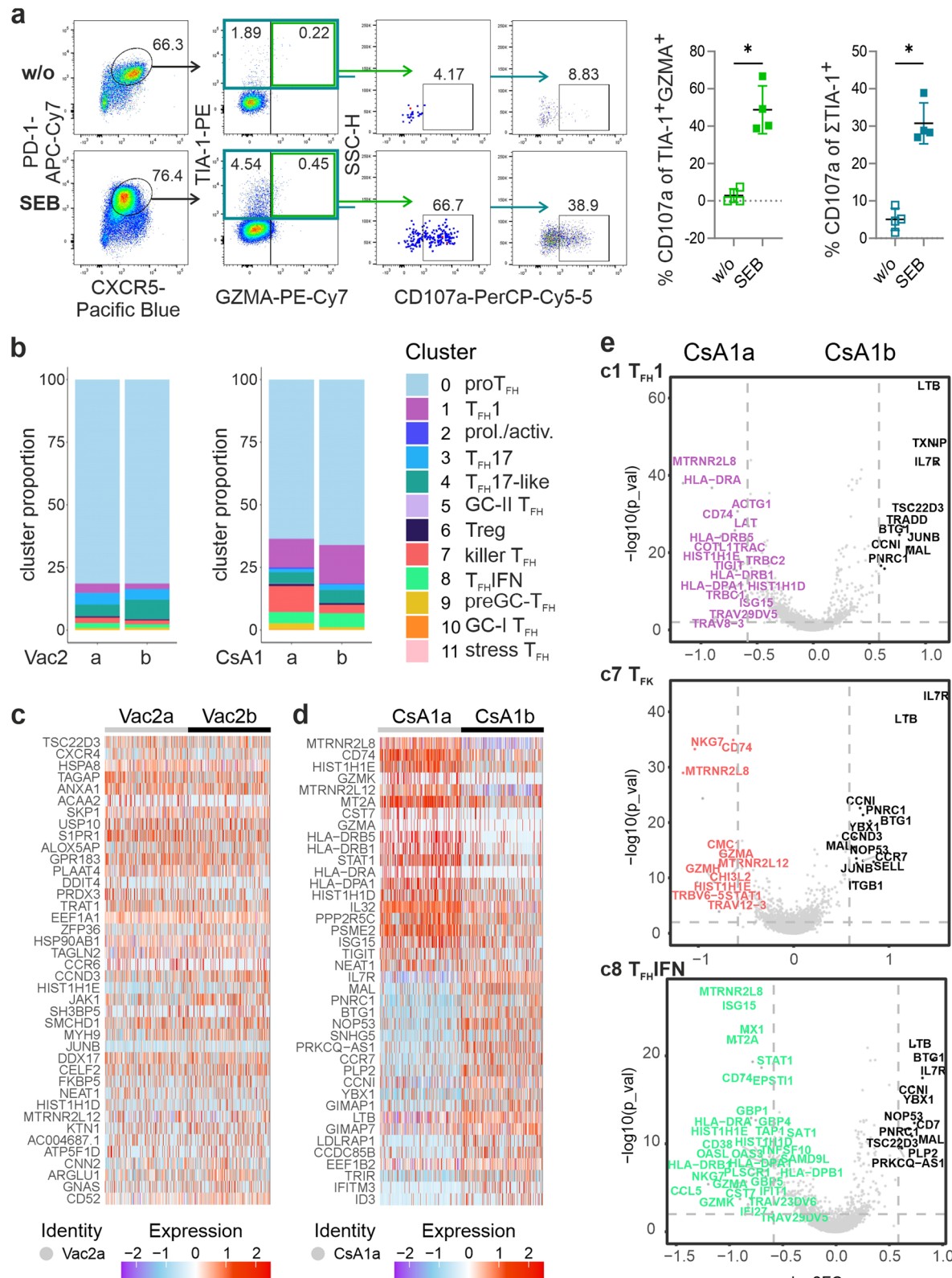

**Fig. 4 | $T_{FH}$ cells from c7 degranulate upon SEB stimulation and diminish upon CsA tapering. a** SEB-stimulated tonsil cells were stained for CD4 (gate) CXCR5$^+$PD-1$^+$ $T_{FH}$ cells after 4 days. Representative dot plots and cumulative graphs are shown for TIA-1$^+$GZMA$^+$CD107a$^+$ (r = −0.821) and all TIA-1$^+$ $T_{FH}$ cells expressing CD107a (r = −0.816) (n = 4 tonsils; Mann–Whitney test, *p ≤ 0.05).

**b** Differences in cell type composition (in percent) between Vac2a vs Vac2b (left) and CsA1a vs CsA1b (right). Heatmap of top 20 differentially expressed genes in Vac2a vs Vac2b (**c**) and CsA1a vs CsA1b (**d**). **e** Volcano plots for c1, c7, and c8 individually showing the impact of CsA tapering (CsA1 vs CsA1b).

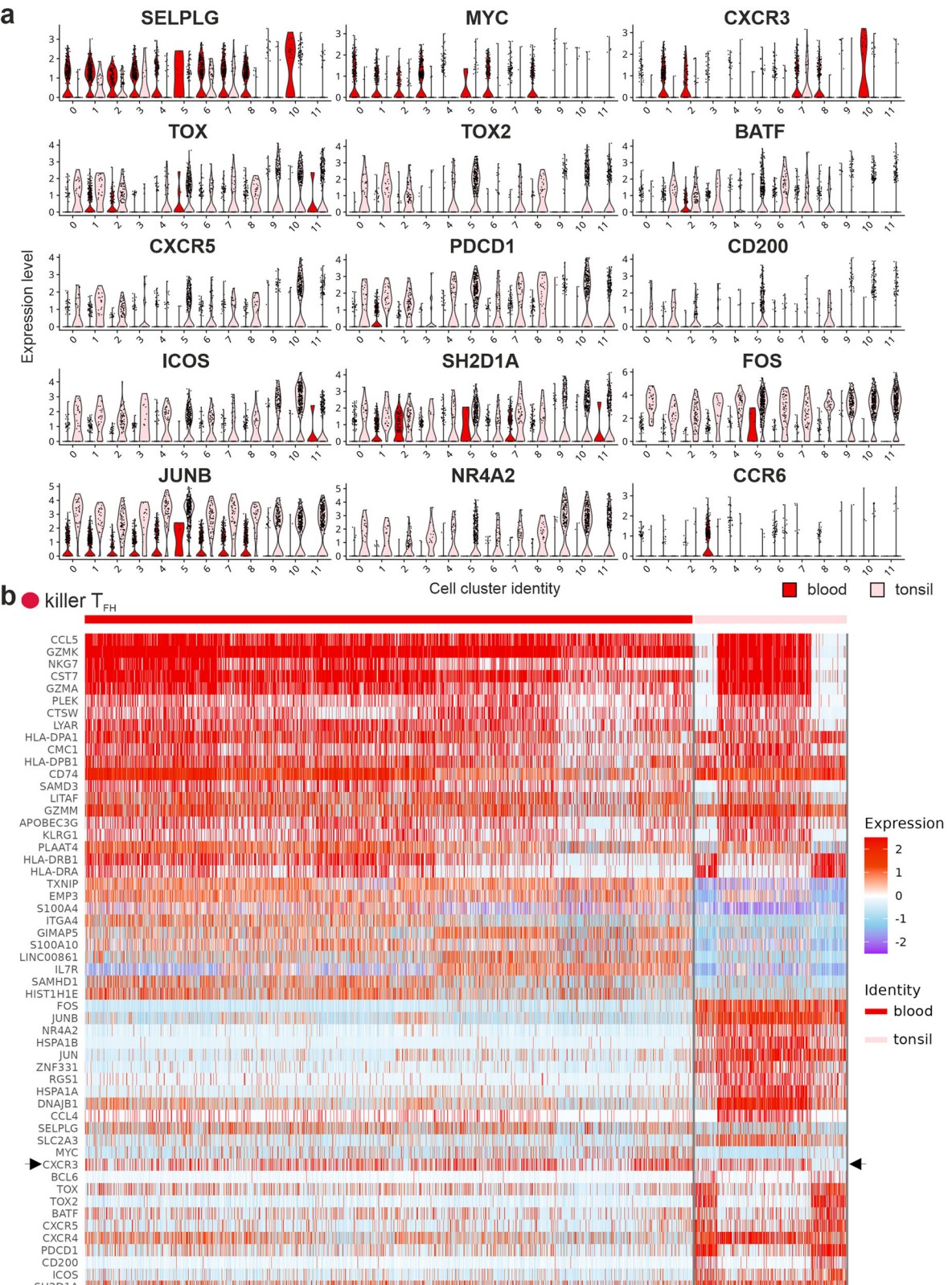

**Fig. 5 | T_FK cells derived from tonsils or PB express *SELPLG*, *CXCR3* and *SH2D1A*. a** Violin plots depicting chosen marker in all clusters, either gathered from PB (red) or tonsils (dusky pink). **b** Heatmap of c7, the upper part with cluster-defining transcripts. The middle part depicts PB-preferential and the lower part tonsil-standard transcripts. The arrow points to *CXCR3*.

showed only weak connections to the other clusters. Nevertheless, c0, which are characterized as proT_FH cells, share TCRs with the classical c5, c9 and c10 within the tonsils, suggesting a progression from c0 to bona fide T_FH cells (Fig. 8b).

In Ton2 and CsA1a, the T_FK cells in c7 appeared most distinct from any other T_FH cell cluster, underscoring their uniqueness among CD4⁺CXCR5⁺ cells (Fig. 8a). Still, the search for shared TCR clones between T_FH sub-populations revealed connections between c7 and all other clusters except c3

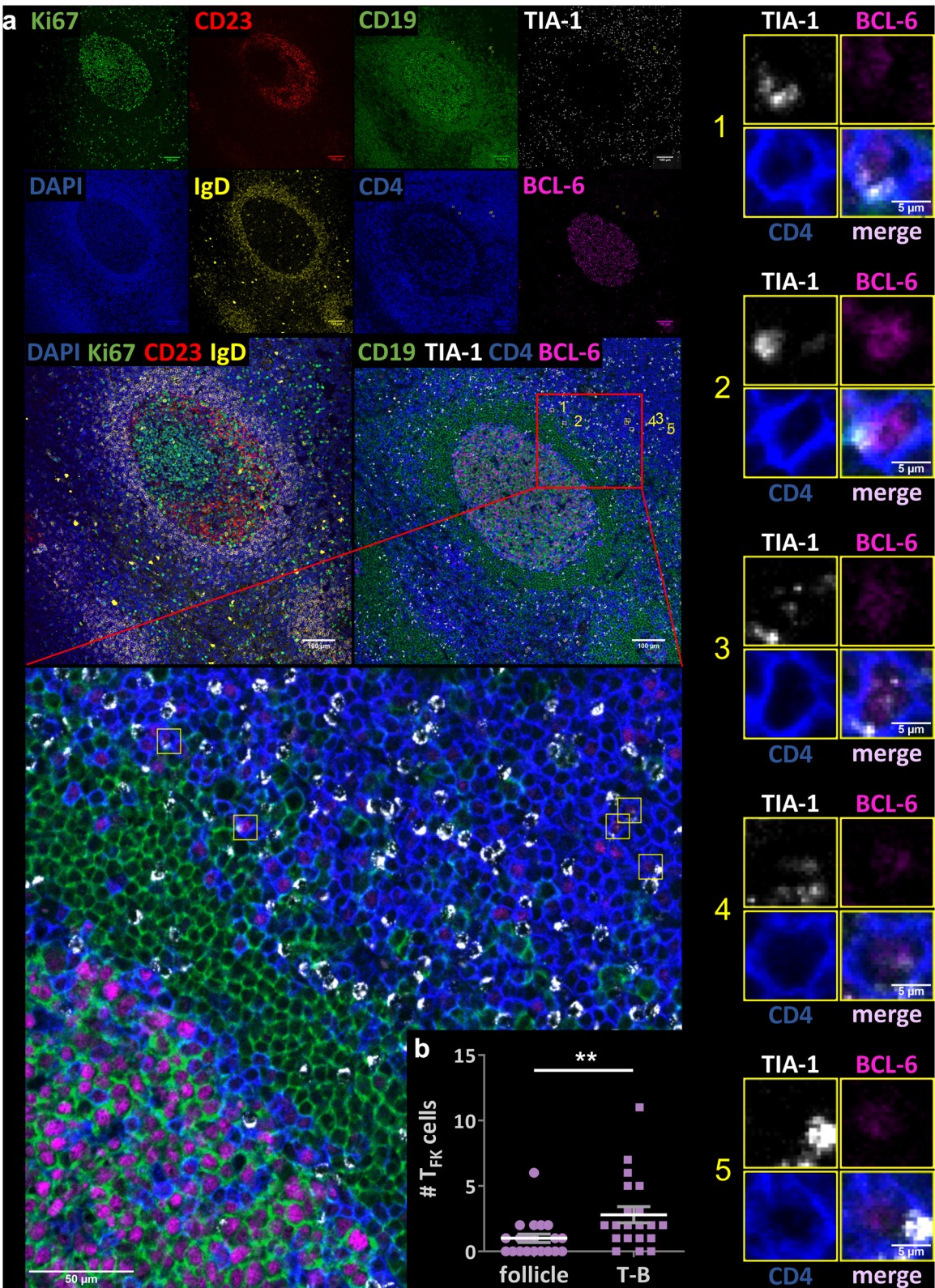

**Fig. 6 | CD4$^+$BCL-6$^+$TIA-1$^+$ T cells reside outside of the GC. a** Human tonsillar FFPE tissue sections from donors (habitual snoring) were stained for TIA-1/NKG7$^+$ T$_{FK}$ cells. A representative image of one section stained with either GC-B cell (Hoechst/blue, Ki67/green, CD23/red and IgD/yellow) or CD19 (green), CD4 (blue), BCL-6 (magenta), and TIA-1 (white; antibody for NKG7) (right panel and enlarged below, scale bars either 100 or 50 μm) is shown (Ton9). Individual pictures of five T$_{FK}$ cells each with anti-CD19, -CD4, -BCL-6, -TIA-1 (alone and merged) are depicted to the right, scale bars = 5 μm. **b** Quantitation of the number of T$_{FK}$ cells at the T-B border *vs* within GCs (follicles/n = 30; Mann–Whitney test, **$p \leq 0.01$, here $p = 0.006$). If images did not reveal any T$_{FK}$ cells in either the T-B border or within the GC or both, this is also represented.

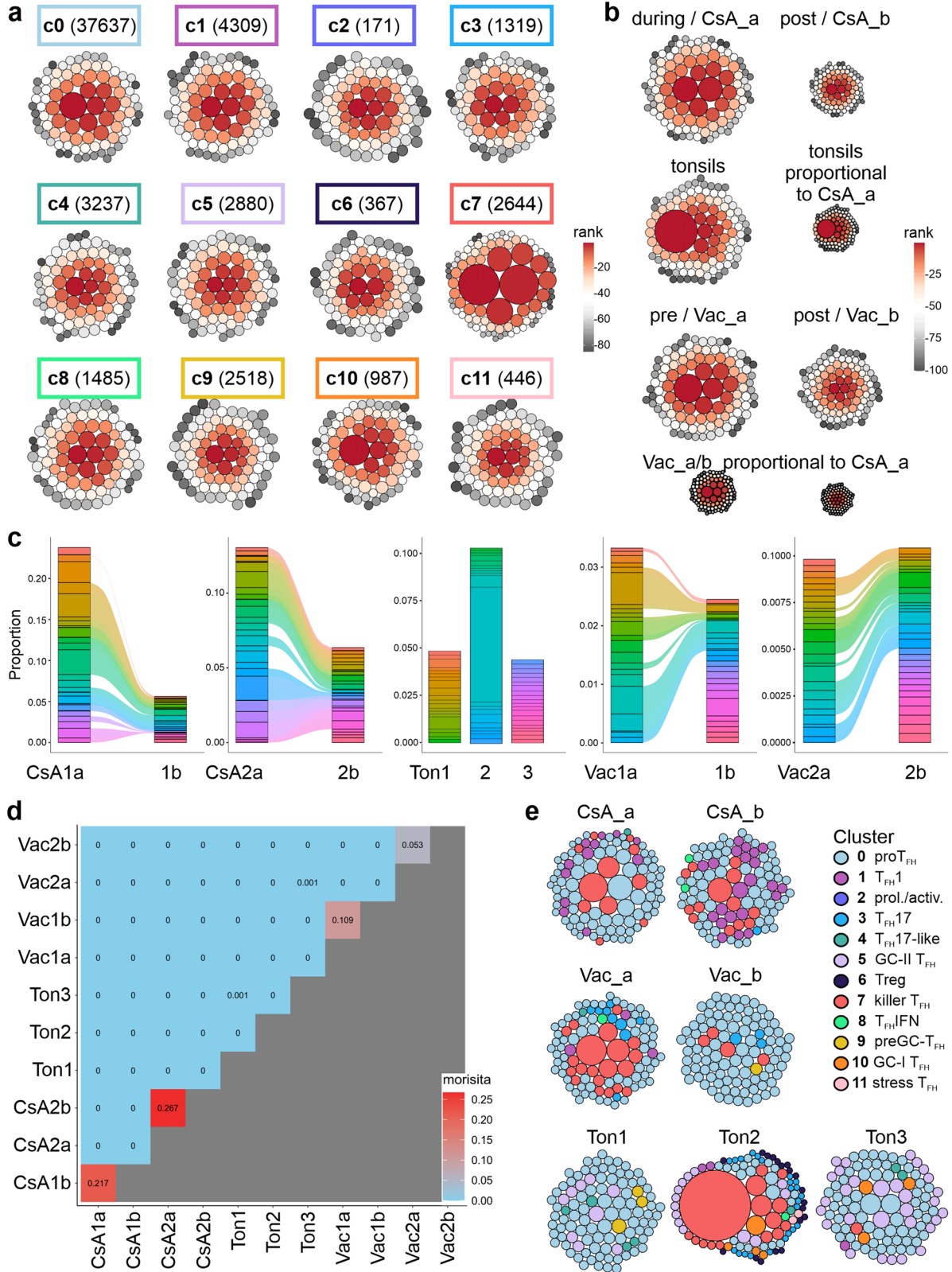

**Fig. 7 | The most abundant TCR clones derive from c7. a** Bubble plots visualize the top 85 clonotypes per cluster. The size of the circles is determined by the frequency of cells representing a clone of the cluster and the color gradient indicates the rank of a clone in frequency (1-85). The box above each plot demonstrates the represented cluster next to the number of included cells in parentheses. **b** Bubble plots visualize the top 100 clonotypes per donor group. The size of the circles is determined by the frequency of cells representing a clone of the group and the color gradient indicates the rank of a clone in frequency (1-100). Circle sizes of the different groups are plotted in proportion to each other. **c** Repertoire analysis reveals changes in diversity. **d** Overlap analysis using Morisita's overlap index. Similarity between each TCR repertoire was measured, red indicates pairwise correlation. **e** Bubble plots visualize the top 100 clonotypes per donor/donor group. The size of the circles is determined by the frequency of cells representing a clone of the donor/donor group and the colors indicate to which cluster each clone belongs.

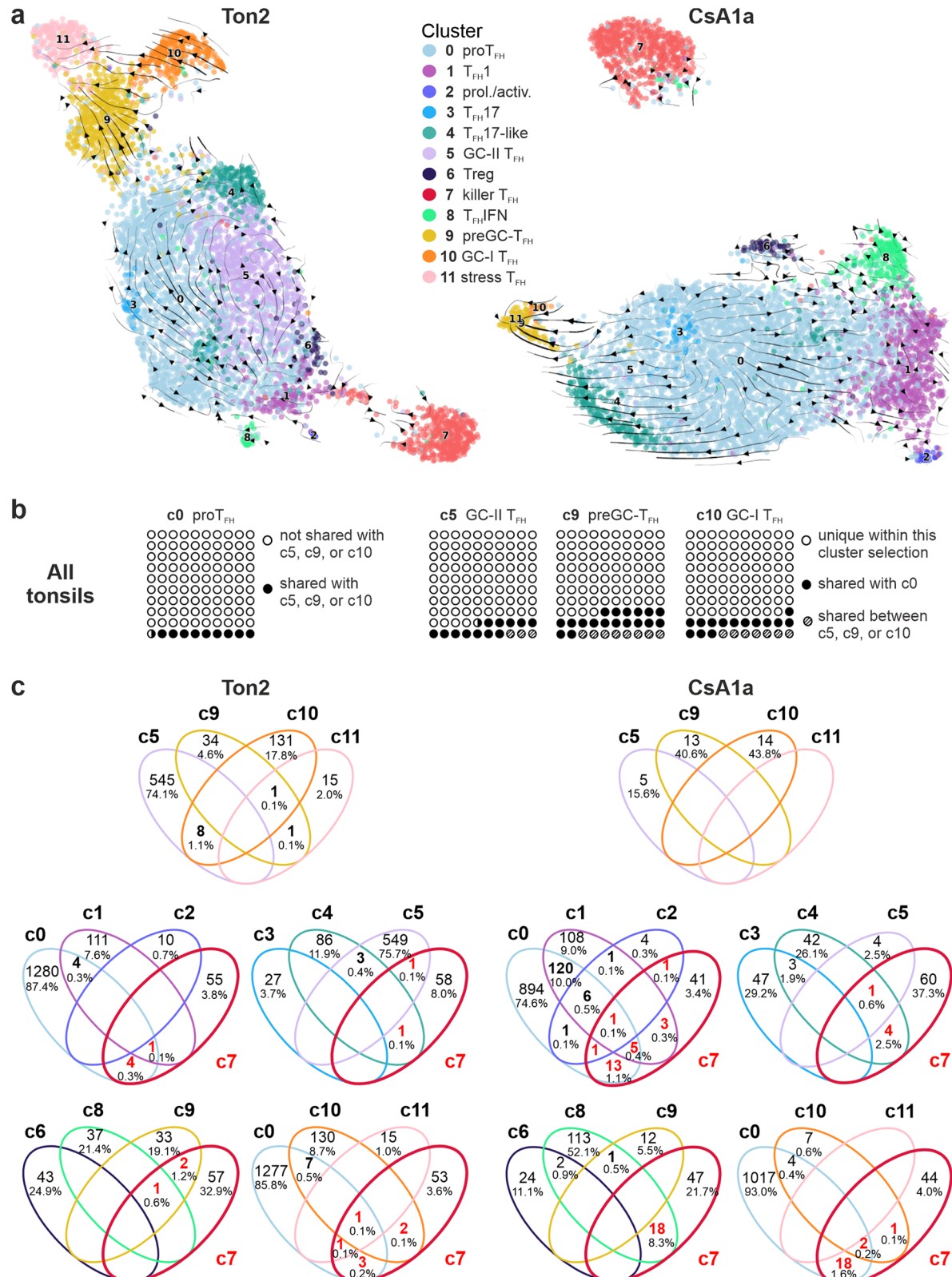

**Fig. 8 | $T_{FK}$ cells share TCRs with other CD4⁺CXCR5⁺ T-cell subpopulations.**
**a** Extrapolated future states are shown as arrows in pre-calculated UMAP for Ton2 and CsA1a. **b** Pie charts of all tonsils combined, where a circle represents 1% of the data within each square / cluster. Filled circles (•) indicate shared TCR clones between c0 and c5, c9, or c10. **c** Venn diagrams present overlapping clonotypes between clusters of Ton2 and CsA1a. The frequencies of unique clonotypes and shared clonotypes between clusters are indicated in the corresponding area. The first row gives the degree of shared TCRs among classical c5, 9-11, while the second and third rows focus on shared clones of c7 with each of the other clusters.

$T_{FH}17$ and c6 Treg. C7 clones were most frequently shared with c0 proT$_{FH}$ in both tissue origins, followed by c8 T$_{FH}$IFN in CsA1 or, remarkably, c9-11 in Ton2 (Fig. 8c). After all, this suggests that CD3$^+$CD4$^+$CD45RA$^-$CXCR5$^+$ cells encompass diverse helper-like and one cytotoxic T$_{FH}$ cell subpopulation, interconnected and related to each other.

## Discussion

CD4$^+$CXCR5$^+$BCL-6$^-$ cT$_{FH}$ cells emerge with varying surface molecules and a certain potential to support B-cell differentiation[9–14,17]. To understand their relationship to GC-T$_{FH}$ cells, we applied scRNAseq to sorted CD3$^+$CD4$^+$CD45RA$^-$CXCR5$^+$ T cells from PB and tonsils. Reassuringly, all peripheral CXCR5$^+$CD4$^+$ subtypes were present in tonsils, too. Thus, although human naive CD4$^+$ T cells transiently upregulate CXCR5 upon activation[4,63], the CXCR5$^+$CD4$^+$ T cells identified represent cT$_{FH}$ subtypes potentially migrating in and out of SLOs.

T$_{FH}$1, T$_{FH}$17, and T$_{FH}$IFN as well as a subtype with a weak classical T$_{FH}$ phenotype (c9), originated from the CXCR5$^+$CD4$^+$ progenitor T$_{FH}$ (c0) in both PB and tonsils. Interestingly, tonsillar proT$_{FH}$ cells expressed the full set of canonical T$_{FH}$ genes, raising the question of whether further T$_{FH}$ differentiation occurs in parallel at both sites or whether proT$_{FH}$ cells are transitory while the tissue enforces site-specificity. T$_{FH}$ cells may lose their bona fide T$_{FH}$ phenotype upon loss of BCL-6, thereby de-repressing CCR7 and CD127[35]. SAP (*SH2D1A*) deficiency, which prevents GC-T$_{FH}$ differentiation, while cT$_{FH}$ cells still appear, implicates the exclusive release of pre-T$_{FH}$ to PB[64]. This is consistent with the described developmental trajectory of T$_{FH}$ and especially T$_{FR}$ cells[65]. Nevertheless, BCL-6 oscillations during all stages prior to an imprinted GC-T$_{FH}$ phenotype could—in a network[2,35]—dictate a pronounced T$_{FH}$ character of all CD3$^+$CD4$^+$CD45RA$^-$CXCR5$^+$ T-cell types within SLOs.

One unifying transcript, upregulated in all tonsillar subclusters was *SLC2A3* (encoding GLUT3), curiously repressed by BCL-6 in T$_H$1 cells[66]. In murine pathogenic T$_H$17 cells, GLUT3 controls a metabolic-transcriptional circuit, which influences their epigenetic landscape[67]. Thus, metabolic regulators do not only discriminate between T$_{FH}$ and T$_H$1 differentiation[68], but could also enforce tissue-specific changes in T$_{FH}$ cells.

To our surprise, we found cells with a cytotoxic phenotype within the CXCR5$^+$CD4$^+$ T-cell pool in PB and tonsils from healthy donors and patients after allo-HSCT, which we termed T$_{FK}$ cells. Cytotoxic T$_{FH}$ subpopulations with perforin and granzyme B expression have been detected among CD4$^+$ T cells in children with group A Streptococcal recurrent pharyngitis (GAS RT) and in severely ill COVID-19 patients among virus-specific CD4$^+$ T cells[69,70]. Tonsils of these patients harbor smaller GC and unveil reduced GCR. In comparison, the cytotoxic phenotype of T$_{FK}$ cells seemed even more pronounced with high expression of several different granzymes, *NKG7, CST7* and *CCL5* controlled by EOMES and BLIMP-1. Of note, the strong cytotoxicity of T$_{FK}$ cells was evident in relation to T$_{FH}$ cells with a helper transcriptome, while it was obvious but minor in comparison to cytotoxic CD4$^+$ T$_H$1 cells in COVID-19 patients[70]. Still, if the phenotype of the described cytotoxic GZMB$^+$ T$_{FH}$ cells and the T$_{FK}$ cells described here would be just a continuum of the same differentiation pathway, it can count as another example of the remarkable plasticity of (cytotoxic) CD4$^+$ T cells strongly influenced by signals from the microenvironment.

CD8$^+$ T cells release the chemokine CCL5 together with perforin and granzyme A during chronic infections[71]. Intriguingly, various authors have reported that during chronic inflammatory conditions, expanding cytolytic CD4$^+$ T cells, which are reprogrammed T-helper cells, keep-up their MHCII restriction, exhibit a terminally differentiated phenotype and may be beneficial to the host due to direct cytotoxicity against infected cells[72–76]. The opposite is true for secondary progressive multiple sclerosis (SPMS), where CNS-resident EOMES$^+$GZMB$^+$NR4A2$^+$CD4$^+$ T cells correlate with active progressive disease[52]. No phenotypic analyses identifying these cells as T$_{FH}$ cells were undertaken, but the existence of T$_{FR}$-empty leptomeningeal follicle-like structures in SPMS together with the finding in mice that the absence of T$_{FR}$ cells provokes the appearance of cytotoxic T$_{FH}$ cells, hints at the possibility[77,78]. Our data that the dominance of T$_{FK}$ cells under

calcineurin inhibition can be lifted after tapering of CsA implies that regular activation of T cells, presumably via their TCR and in an acute context, could counteract chronic manifestations.

The T$_{FK}$ cells were oligoclonal, possibly driven by chronic infections such as EBV or CMV, as oligoclonality is most prevalent in memory inflation of CD8$^+$ T cells[79]. As we sorted for CD45RA$^-$ cells, T$_{EMRA}$ cells were excluded by definition, yet high CX3CR1 and KLRG1 expression indicated terminal differentiation. In tonsils, T$_{FK}$ cells located outside GCs, in some aspects resembling extrafollicular CXCR5$^{hi}$PD-1$^{hi}$ T$_{FH}$ cells with low BCL-6, but upregulated PSGL1, CXCR3 and BLIMP-1[80]. Of note, GC-T$_{FH}$ cells partly phenocopy pre-exhausted CD8$^+$ T (T$_{PEX}$) cells[41,81]. The cytotoxic T$_{FK}$ cells are even closer to this common T$_{FH}$/T$_{PEX}$ signature, implicating a shared signaling pathway upon chronic activation, at the same time terminal differentiation similar to the proficient antiviral T$_{EMRA}$ cells. With their MHCII restriction, T$_{FK}$ cells may represent a type of APC-killing regulatory cell recruiting further CCR5$^+$ T$_{FK}$ cells, myeloid and NK cells through the chemokines CCL3, 4 and 5[82], thus balancing the humoral immune response in the vicinity of GCs.

PBMCs reflect their tissue-based counterparts only poorly, which may hold true for T$_{FH}$ cells prone to stay within the SLO or even home to a GC[65,83–85]. Still, our data indicate that cT$_{FH}$ cells – and in particular the newly defined subpopulation of cytotoxic T$_{FK}$ cells – share subset-specific phenotypes with tonsillar equivalents and thus may provide helpful parameters in clinical situations.

## Methods

### Patient and healthy donor samples

Tonsils were collected from donors with no, mild, and recurrent / moderate tonsillitis (Supplementary Table 1). cT$_{FH}$ were isolated from PBMCs of healthy volunteers ($n = 13$), from healthy individuals before and after medically indicated routine booster vaccination (tetanus toxoid; $n = 2$) and (from PBMCs) from patients before and after tapering of cyclosporin A (CsA; $n = 2$) following allogeneic hematopoietic stem cell transplantation (allo-HSCT). For the Study Design, see Supplementary Fig. 1a. For all, informed consent was obtained after explanation of the nature and possible consequences of the studies, beforehand approved from the institutional ethics committee (Ethik-Kommission der Friedrich-Alexander-Universität Erlangen-Nürnberg). All ethical regulations relevant to human research participants were followed.

### Flow cytometric cell sorting for transcriptome and TCR repertoire analysis

Cryopreserved samples were rapidly thawed in a water bath at 37 °C, diluted to 10 ml with warm RPMI 1640 (BioWhittaker, Lonza, Walkersville, MD, USA)/10% FCS/10 mM HEPES, centrifuged ($300 \times g$, 10 min) and either counted (tonsils) or directly resuspended in 100 μl FACS buffer (PBS/2% FCS). Cells (approx. $5 \times 10^6$) were stained with fluorochrome- and oligo-coupled antibodies (Supplementary Table 2) for 20 min at 4 °C, washed once and resuspended in 900 μl FACS buffer. Immediately prior to sorting, DAPI (Sigma-Aldrich, Merck, Darmstadt, Germany) was added for dead cell exclusion. Sorting was performed on a FACSAria™ Fusion high-speed cell sorter (4-way purity; 85 μm nozzle; BD Biosciences); gating strategy in Supplementary Fig. 1b, c. Cells were sorted into 1.5 ml DNA LoBind tubes (Eppendorf, Hamburg, Germany) prefilled with 0.5 ml PBS/10% FCS and directly processed for molecular analysis.

### scRNAseq, scTCRseq and scCITEseq library preparation and sequencing

Freshly sorted human T$_{FH}$ cells (between 11,000 and 20,000 per sample) were loaded onto the Chromium Single Cell Controller (10x Genomics) using the Single Cell 5′ Library & Gel Bead Kit v1.1 (10x Genomics #PN-1000165). cDNAs were amplified using 13-14 cycles of PCR. ScRNAseq libraries were constructed using 14-15 cycles of PCR and scCITEseq libraries were constructed using 9 cycles of PCR. ScTCRseq libraries were generated from the same cDNAs using the Single Cell V(D)J Enrichment

Kit, Human T Cell (10x Genomics) (10 cycles of PCR). Products were purified using Ampure XP beads and quality was controlled using Agilent Tapestation. ScRNA- and scCITEseq libraries were paired-end sequenced (S1 flow cell, 100 bp) on the Illumina NovaSeq 6000™. ScTCRseq libraries were single end sequenced (High Output flow cell, 150 cycles) on the Illumina NextSeq 550.

## Single-cell transcriptome profiling

Cellranger 6.1 was used to align the sequencing reads to a reference human genome to generate the counting matrix. Data analysis was performed using Seurat 4[86], GitHub: https://github.com/satijalab/seurat. Low quality cells defined by high levels of mitochondrial gene expression (>20%) were filtered out, in addition, only cells of gene feature numbers ranging between 300 and 3500 were kept for subsequent analysis. Here we excluded one gene feature, XIST to avoid the bias introduced by the donor sex. Batch effects were initially removed with the Seurat's integrated canonical correlation analysis (CCA). Anchor genes were computed, which enabled precise data integration and cell clustering. Afterwards cell clusters were identified by the Seurat function FindCluster, where a resolution parameter 0.5 was specified and the Louvain algorithm was applied. Cluster annotation was carried out with SingleR (version: 1.8, GitHub: https://github.com/LTLA/SingleR) package[87] and PanglaoDB[88]. ADT de-multiplexing and analysis were performed in Seurat, normalization method CLR. Non-linear dimension reduction for visualization was achieved by UMAP.

## Single-cell TCR repertoire profiling

The cellranger vdj pipeline (version 6.1) was applied to count T-cell receptor sequencing data produced by VDJ libraries to obtain the profile, followed by data analysis using R scRepertoire package[89], GitHub: https://github.com/ncborcherding/scRepertoire. For the generation of bubble plots, TCR clonotypes were defined as cells expressing the same CDR3 nucleotide sequence in their TCRα and TCRβ chains, respectively. Cells lacking the combined CDR3 nucleotide sequence information were excluded from the analysis. For direct comparisons of clonality between groups, the number of cells considered per group was randomly reduced to the cell number of the smallest group. TCR clonality analysis was performed in *R* and visualized by using the packages *packcircles*, GitHub: https://github.com/mbedward/packcircles, and *ggplot2*, GitHub: https://github.com/tidyverse/ggplot2.

Morisita's overlap index measures how many times two sampled points are more likely to be randomly selected from the same quadrat (the dataset is covered by a regular grid of varying size) than they would be with a random distribution generated by a Poisson process. Duplicate objects are merged, and their counts are summed. Sample overlap in this study is estimated using Morisita's overlap index.

## RNA velocity analysis

Splicing information was collected with velocyto, GitHub: https://github.com/velocyto-team/velocyto.py, examining all the bam files generated by cellranger. The results were imported into scVelo for downstream RNA-velocity analysis[90]. The figure was plotted in a grid and stream style implemented in the scVelo, GitHub: https://github.com/theislab/scvelo.

## Flow cytometric analysis of human MNC from peripheral blood and tonsils

PBMCs were isolated from freshly drawn blood samples by density gradient centrifugation over Ficoll (Pancoll human, PAN BIOTECH, Aidenbach, Germany). Frozen tonsil MNC were rapidly thawed (see above). Staining was performed in PBS/2% FCS ($1 \times 10^6$ cells/100 µl) and brilliant stain buffer (BD Biosciences, Heidelberg, Germany). For dead cell exclusion, fixable viability dye (Zombie UV™, BioLegend, San Diego, CA, USA) was added prior to staining. For staining of intracellular markers, the eBioscience Foxp3/Transcription Factor Staining Buffer Set (Invitrogen by Thermo Fisher Scientific, Carlsbad, CA, USA) was used. Data were acquired on a FACSymphony™ A5 (BD Biosciences) and analyzed with FlowJo® v10.8.1

(Treestar Inc., Ashland, OR, USA). Antibodies used for staining are listed in Supplementary Table 3.

## Immunofluorescence (IF) histology staining and analysis

Consecutive formalin-fixed paraffin-embedded (FFPE) sections were used to localize $T_{FK}$ cells. Deparaffinized FFPE sections, 3-4 µm from each embedded block, underwent heat-induced antigen retrieval (20 mM citric acid buffer, pH 6.0). Sections were blocked with Antibody Diluent (Dako, #S3022) for 1 h at RT before incubation with primary antibodies (Supplementary Table 4) in Antibody Diluent for 1 h. For thoroughly washed sections (3x with TBST) secondary antibodies (1:1000) were prepared in PBS containing 0.05% Tween 20 (Sigma, P9416-50ML) and Hoechst for 1 h at RT. After washing with TBST, sections were embedded in Fluoromount-G Mounting medium (ThermoFisher, #00-4958-02). Representative images were acquired at confocal laser-scanning microscope Zeiss LSM780. Plan-Apochromat 20 × 0,5 and C-Apochromat 40 × 1.2 W objectives were used for detection in four simultaneous channels.

Image analysis was performed with Fiji (*Fiji is Just ImageJ*)[91], for which a macro was designed to incorporate both deconvolution and segmentation for the images acquired with the confocal microscope[92]. In a next step, cell counting for B cells, CD4$^+$ T cells, and BCL-6$^+$ cells as well as for TIA$^+$ cells was achieved using a pipeline created in CellProfiler[93], before a second Fiji macro showed the overlap between the segmented images to indicate where $T_{FK}$ cells may be located. To confirm the overlap of the three markers CD4, BCL-6 and TIA-1 in one single cell, intensity profiles along the cell width for the likely cell candidates were created by a third Fiji Macro. For the intensity profiles to confirm the cell candidate as a $T_{FK}$ cell, CD4 had to be expressed at the cell membrane and overlap with TIA while BCL-6 had to be expressed in the nucleus.

## In vitro stimulation of adenoid cultures

Tonsil cells were thawed and resuspended with complete medium (RPMI with Glutamax, 10% FBS, 1× non-essential amino acids, 1× sodium pyruvate, 1× penicillin-streptomycin, 1× primocin, 50 mM β-mercaptoethanol). For cell stimulation, adenoid cells were plated in 96-well-U-plate, 2*10^5 cells/well (1*10^6 /ml) and stimulated with 1 µg/ml of staphylococcal enterotoxin B (SEB; Sigma S4881-1MG) for 4 days. Afterwards, cells were harvested for flow cytometry.

## Reproducibility and statistics

The number of samples used is indicated in the figure legends. Multiple independent studies confirmed consistent results. Sequencing data were analyzed using Seurat 4, scRepertoire, ggplot2, velocyto, scVelo, packcircles, and R as outlined in the respective paragraphs under "Methods". The in vitro experiment was performed on four different donors and repeated once independently; the ex vivo experiments were performed on several donors each time as specified in the figure legend, i.e. all measurements were taken from distinct samples. Quantitation of flow and histological data are shown as the mean ± SEM and were analyzed using two-way ANOVA with Šídák post-hoc analysis or Mann–Whitney test as indicated in the figure legends. *P* values less than 0.05 were considered significant.

## Reporting summary

Further information on research design is available in the Nature Portfolio Reporting Summary linked to this article.

## Data availability

Sequencing read counts after mapping and filtering are deposited at GEO (accession number GSE218131) for scRNAseq, CITEseq and TCRseq. Due to privacy and ethical concerns, the raw sequencing data cannot be made available. The European data protection rules (GDPR) consider sequencing data of human individuals as private and requiring special protection. Unfortunately, the original consent did not explicitly include the sequencing of biomaterial, and hence we are not allowed to publish, release, or even share the raw data. The ethics vote is independent of data protection issues.

If not included in the article, data are uploaded as supplementary information. Supplementary Data 1 lists the first 50 differentially expressed genes of each cluster. Supplementary Data 2 give the numerical sources for the Figs. 2c, 3d, 4a, 6b and Supplementary Fig. 3d.

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

## Acknowledgements

We are indebted to all members of the FACS and NGS core facilities at the LIT for excellent experimental assistance. We further thank Andrea Schneider and Mareike Massing, Erlangen, for expert technical support. This work was supported by the Deutsche Forschungsgemeinschaft (DFG, German Research Foundation), project number 324392634 - TRR 221, seed funding (S.Sp., M.R., P.H., T.H.W., F.B.-S.), B01 (S.Sp., F.B.-S.), B07 (M.E. M.R., P.H.), B10 (J.W., T.W.), and Z01 (A.R.). Additional funding was received from the DFG /FOR 2830 (F.B.-S.) and DFG / BE2309/8-1 (F.B.-S.). D.D. acknowledges DFG, DU548/5-1 420943261.

## Author contributions

C.L. and N.S. performed bioinformatic analyses. S.Sp. and I.S. performed research and together with J.W., C.A. and D.D. provided clinical samples. Y.X. and S.S.H. performed and analyzed IF histological staining, K.M.H. examined TCR clonality; A.R. and M.E. supervised experiments and provided resources; S.Sp., M.R., P.H., T.H.W. and F.B.-S. conceived the project, designed and organized the research and interpreted data. R.E. and P.H. performed cell sorting, while H.S. prepared samples for sequencing. F.B.-S. drafted the manuscript.

## Funding

## Competing interests

The authors declare no competing interests.

## Additional information

[1]Functional Genomics and Systems Biology Group, Department of Bioinformatics, Biocenter, Julius-Maximilians-University Würzburg, Würzburg, Germany. [2]Department of Internal Medicine 5, Hematology/Oncology, University Hospital Erlangen, Friedrich-Alexander-University Erlangen-Nürnberg (FAU), Comprehensive Cancer Center Erlangen-EMN, Erlangen, Germany. [3]Institute of Pathology, Julius-Maximilians-University Würzburg, Würzburg, Germany. [4]Division of Genetics, Department Biology, Nikolaus-Fiebiger-Center of Molecular Medicine, Friedrich-Alexander University Erlangen-Nürnberg, Erlangen, Germany. [5]Leibniz Institute for Immunotherapy, Regensburg, Germany. [6]Department of Internal Medicine III, University Hospital Regensburg, Regensburg, Germany. [7]Department of Otorhinolaryngology, Head & Neck Surgery, Else Kröner-Fresenius-Foundation-Professorship, Section of Experimental Oncology & Nanomedicine (SEON), University Hospital Erlangen, Friedrich-Alexander University Erlangen-Nürnberg, Erlangen, Germany. [8]Laboratory of Dendritic Cell Biology, Department of Dermatology, University Hospital Erlangen, Friedrich-Alexander University of Erlangen-Nürnberg, Erlangen, Germany. [9]Comprehensive Cancer Centre Mainfranken, Julius-Maximilians-University of Würzburg, Würzburg, Germany. [10]Present address: Institute of Immunology, Jena University Hospital, Friedrich-Schiller-University, Jena, Germany. [11]These authors jointly supervised this work: Petra Hoffmann, Thomas H. Winkler, Friederike Berberich-Siebelt. path230@mail.uni-wuerzburg.de

