## [Peer Review File · Communications Biology]

Reviewers' comments:

Reviewer #1 (Remarks to the Author):

The authors demonstrated the relationship of CD4+CXCR5+ Tfh cells between secondary lymphoid tissue (tonsil) and peripheral blood using scRNAseq. CD4+CXCR5+ Tfh cells in both shared subclusters, especially cluster 7, which was named Tfk cells here and expressed TIA-1, CXCR3, and GZMB. The scTCR repertoire analysis showed that Tfk cells shared TCRs with other subclusters. Therefore, the authors concluded that Tfk cells represent terminally differentiated cells within Tfh cell subsets, and cTfh cells share phenotypes with their tonsillar equivalents. The results of this study are of high interest to immunologists and clinicians because they could have applications in vaccination. However, unfortunately, the paper seems to lack data to support the authors' conclusions, as one gets the impression that data analysis based on scRNA analysis and the evidence of previous reports has not been fully integrated and adequately described.

Specific comments

(1) In scRNAseq data integration, it seems that cluster 0 needs sub-clustering. In Fig. 1d, the feature map of CCR7, TOX, MYC, S1PR1, and IL-21 shows heterogeneous expression within cluster 0. The authors probably need to identify the optimal FindCluster resolution or explore alternative clustering packages (<https://github.com/lanevskiAleksandr/sc-type>).

(2) Both Fig. 1d and Fig. 2a should include annotations for expression levels (indicated by a yellow-to-red heatmap).

(3) In Fig. 2c, the detection levels of GZMB vary between PBMC and Tonsil. These FACS plots suggest unequal detection. These plots should be reconsidered.

(4) Fig. 4 and Supplementary Fig. 4 depict the relationships of CD4+CXCR5+ Tfh cells between the tonsil and peripheral blood. In these results, is it possible to align data from the same patient treated with tonsillectomy? It might be better to integrate the Ton1/2/3 and Vac1/2a data without the inclusion of pathological allo-HSCT data.

(5) The authors suggest that Tfk cells are terminally differentiated from classical Tfh cells in Ton2 and CsA1a data. However, RNA velocity in Fig. 7a suggests that cluster 7 is isolated. The authors might consider another trajectory analysis (e.g., Monocle) to improve the isolation of cluster 7.

(6) It's worth noting that Tfk cells might not be entirely novel, as GZMB+ cytotoxic Tfh cells have already been reported (Ref 60, 61). If the authors intend to differentiate Tfk cells from GZMB+ cytotoxic Tfh cells, more validation experiments, functional assays, and correlation studies on Tfk cells and their pathophysiology are needed.

(7) In the methods section, it would be beneficial to include the GitHub URL for the analysis package.

(8) The authors should accurately reference their sources (e.g., Ref 52, Ref 60).

(9) Some sentences lack clear subjects or have none. Please provide the necessary clarifications.

Reviewer #2 (Remarks to the Author):

Summary of research:

The authors study human peripheral blood (PB) and tonsil CD3⁺ CD4⁺ CD45RA⁻ CXCR5⁺ TFH cells by integrative 5' scRNA-seq analysis. They build a dataset of single-cell gene expression, TCR-seq, and CITE-seq of a handful of surface markers for a total of approximately 70,000 cells sorted from 11 distinct tonsil (n=3) or PBMC (n=8) samples. PBMC samples include blood drawn from healthy donors (n=2) before and 7 days after booster vaccine injection, or from allo-HSCT transplant recipients (n=2) before and after cyclosporin A tapering. The authors use a classic single-cell data analysis workflow based on the Seurat analysis tool for the gene expression data, and the scRepertoire analysis tool for the TCR-seq data. After batch integration, the authors find 12 gene expression-based clusters of tonsil and PB TFH cells, which they annotate based on marker genes identity and supervised analysis of key markers of previously known TFH subsets. Most of the clusters are composed of cells from tonsils and PB; three clusters (c5, c10, c11; annotated as GC-II TFH, GC-I TFH, and stress TFH, respectively) are composed only of tonsillar cells. The authors also identify that in clusters shared between tonsil and PB cells, cells from tonsils express higher levels of canonical TFH marker genes, whereas cells from PB express more SELPLG, MYC, CXCR3. One of the TFH clusters (c7, annotated as killer TFH or TFK cells) expresses markers associated with cytotoxic T cells (NKG7, CCL5, various granzymes) and is found equally in tonsils and blood. The authors focus on that TFK subset to confirm its surface phenotype and tissue localization based on the expression of the NKG7/TIA-1 marker: TFK cells express PD1 and CXCR3, and locate outside of GC. Analysis of single-cell TCR-seq reveal that TFK cells are oligoclonal, suggesting clonal expansions in response to some antigen. Analysis of shared TCR clonotypes between TFH clusters within samples reveals that TFK cells share a common origin with other TFH subsets.

Assessment:

The study addresses an important question: whether circulating PB TFH cells are good proxies for analyzing TFH responses occurring in secondary lymphoid organs, here tonsils. The results of the study are interesting, based on classic analysis pipelines for that type of single-cell gene expression and TCR-seq data. Thus the results of the single-cell analysis, although descriptive by nature, are robust and of significant interest. The focus on TFK cells is interesting because that subset has been reported occasionally but rarely studied in depth.

Overall, because this is the most original finding, it would be important strengthen the manuscript by providing:

- (i) a quantitative analysis of IF images for robust unbiased TFK localization relative to germinal centers and primary B cell follicles,
- (ii) more insight into the cytolytic function of TFK cells in ex vivo assays
- (iii) a deeper analysis of the impact of vaccine boost or cyclosporin A tapering on the numbers and gene expression profile of those cells.

I also list more specific comments that may be used to enhance the manuscript.

Specific comments:

1. L53 (abstract): the last sentence of the abstract too broad in my opinion when mentioning “counterparts in tissues”. The authors should be more specific: “tonsils”.
2. L68: “High PD-1 expression prevents GC-TFH cell proliferation allowing competition between GC-B cells”: missing reference. This concept is not mentioned in refs 6 or 7.
3. L71-83 (introduction): the authors may also mention T peripheral helper cells (TPH) that have a CD4⁺ CXCR5⁻ PD1⁺ phenotype in blood, have B-helper functions, and have been implicated in a

number of autoimmune diseases in humans.

4. L89-90 (introduction): TFR cells may also derive from TFH cells (iTFR, as reported by Le Coz et al. Science Immunology 2023).

5. L112-113: the authors should rephrase the statement about the advantage that studying PB before and after CsA tapering brings. As such, this is not understandable.

6. L114: define TEMRA.

7. L115 (results): the authors should mention in the results section that they performed batch integration as part of their analytic workflow.

8. L136: it seems that most MYC-expressing cells are not actively proliferating in the dataset.

9. Cluster annotation: there is little detail about how the clusters were named or how the selected genes confirm cluster identification (eg. how was cluster 5 assigned a GC-Tfh label?). It would be interesting to investigate differentially expressed genes between GC-I and GC-II clusters.

10. L142: precise “type 1 interferon response”.

11. L144: define “a classical TFH gene expression pattern”.

12. L212: “...T cells derived from tonsils were more TFH-like than their cTFH counterparts”. The authors are analyzing only TFH-phenotype cells, so the “TFH-likedness” concept is a bit blurry and not properly defined here. It should be properly defined that those genes are part of the canonical TFH marker genes, as observed in other studies of human TFH cells (King et al. Science Immunology 2021, Le Coz et al. Science Immunology 2023).

13. L226-227: there is a discrepancy between the statement “but never within a GC” in the text, and the title of Supplementary Figure 6 (“only some TFK cells locate at/within GC”). A broad quantitative analysis of TFK-phenotype cells in IF images from several tonsil sections of different human donors is required to conclude.

14. L231: what is the quantitative measure of the “highest degree of clonality”? The term “clonality” is ambiguous as it could refer to clonal size, clonal diversity, clonal relatedness, etc.

15. L247: “c0 proved to be the proTfh cells for the classical c5, c9 and c10”. TCR overlap does not provide information on the directionality of the relationships between clusters.

16. L266: “proTFH expressed the entire set of classical TFH genes, questioning whether Tfh differentiation occurs in parallel at both sites”. The phrasing is quite confusing. According to several publications (eg. He et al Immunity 2013 ref 55), precursor TFH emerge in the T zone and some of them reach the efferent lymph and bloodstream, thus forming an early memory pool in the bloodstream, while the others reach the T:B border for full differentiation. Does this dataset question these results or provide more consistent ones?

17. L298: in order to see whether the TFK clones may be specific for EBV or CMV antigens, the authors could try mapping their TCR CDR3 sequences to public repositories of antigen-specific TCR sequences (www.iedb.org).

18. Fig1d: color legend is missing.

19. Fig2a: color legend is missing.

20. Fig3b: color legend is missing.

21. Fig3c: Why are the authors not comparing between CXCR5+TIA+ and CXCR5-TIA-? Are CXCR5 levels continuous or bimodal in CD4+TIA1+ cells?

22. Fig6a: please include C1 instead of 1 etc. Otherwise it looks like a ratio. What does the 2nd number correspond to?

23. Fig6d: define Morisita index.

24. Fig6e: colors in bubble plots and legend do not match (killer TFH notably).
25. Fig7c: those series of Venn diagrams are very hard to read. The authors could use a heatmap display for clonotype overlap, as was used in King et al. Science Immunology 2021.
26. Supplementary Fig3: the authors should include horizontal bars color coded with sample origin (PB or tonsil, sample ID).

Pierre Milpied, Centre d'Immunologie de Marseille-Luminy (CIML)

Reviewer #3 (Remarks to the Author):

Oligoclonal CD4+CXCR5+ T cells with a cytotoxic phenotype appear in tonsils and blood.

Summary:

The authors examined subsets of CD4+CXCR5+ T follicular helper cells (TFH) in the peripheral blood and tonsils to determine whether peripheral blood can be used as a mirror of the TFH cell subsets in lymphoid tissues. Extensive analysis included examination of transcriptomes, TCR repertoire, cell surface expression using single-cell RNA sequencing (scRNAseq) and flow cytometry of sorted CD3+CD4+ CD45RA- CXCR5+ cells. Immunohistochemistry was also performed.

The peripheral blood of 2 healthy individuals was examined before and after vaccination. Tonsils were obtained from 11 individuals with moderate tonsillitis. Three tonsils were extensively examined with scRNAseq, the other 8 were used for Flow Cytometry and histology. In addition peripheral blood was examined from 2 individuals after allogeneic hematopoietic stem cell transplantation before and after the decreasing cyclosporin. Blood from 15 healthy donors was used for flow cytometry alone.

The results of these studies show that similar subsets of CD4+CXCR5+ cells are present in the peripheral blood and tonsils including an additional subset with a cytotoxic phenotype (Tfk). Three subsets of CD4+CXCR5+ cells with T follicular gene expression were only present in tonsils.

Overall the data presented provide strong evidence that a subset of TFH with a killer phenotype Tfk is present in tonsils (outside the germinal center) and in the peripheral blood.

Comments for the authors:

- The experimental design and sorting strategies to obtain TFH are depicted clearly in supplemental Figure 1.
- It would be helpful to show flow cytometry data on PBMC from the 15 healthy volunteers up front. They are now “hidden” in the Figures 2 and 3 and only results of 10 PBMC are shown in Figure 3.
- Extensive characterization using single-cell RNA sequencing (scRNAseq) and flow cytometry of sorted CD3+CD4+ CD45RA- CXCR5+ cells was done on a few normal PBMC and tonsils as well as PBMC obtained after vaccination, and tapering cyclosporine in recipients of allogeneic hematopoietic stem cell transplantation. It is therefore difficult to draw general conclusions based on the results of these experiments.

- Suggestions for clarifying the Figures

Figures:

page 17/39, line 337. Introduce the number of cells loaded on 10x genomics.

page 27/39, line 677. Introduce a color bar for Fig 1a.

page 28/39 line 678. Introduce a color bar for Fig 2a.

- Supplementary Information:

page 6/19 Unable to read the Supplementary Fig 2, specially Supplementary Fig2.a,b,c,and e.

Page 8/19, line 43. Supplementary Fig 3. Unable to read the letter of the x-axis.

Page 10/19, line 51 Supplementary Fig 4a. Unable to read the letter of the y-axis.

Page 4/19, line 70 Supplementary Fig 6b is cut.

Reviewer #4 (Remarks to the Author):

Communications Biology-October 2023

Oligoclonal CD4+CXCR5+ T cells with a cytotoxic phenotype appear in tonsils and blood.

Summary:

The authors examined subsets of CD4+CXCR5+ T follicular helper cells (TFH) in the peripheral blood and tonsils to determine whether peripheral blood can be used as a mirror of the TFH cell subsets in lymphoid tissues. Extensive analysis included examination of transcriptomes, TCR repertoire, cell surface expression using single-cell RNA sequencing (scRNAseq) and flow cytometry of sorted CD3+CD4+ CD45RA- CXCR5+ cells. Immunohistochemistry was also performed.

The peripheral blood of 2 healthy individuals was examined before and after vaccination. Tonsils were obtained from 11 individuals with moderate tonsillitis. Three tonsils were extensively examined with scRNAseq, the other 8 were used for Flow Cytometry and histology. In addition peripheral blood was examined from 2 individuals after allogeneic hematopoietic stem cell transplantation before and after the decreasing cyclosporin. Blood from 15 healthy donors was used for flow cytometry alone.

The results of these studies show that similar subsets of CD4+CXCR5+ cells are present in the peripheral blood and tonsils including an additional subset with a cytotoxic phenotype (Tfk). Three subsets of CD4+CXCR5+ cells with T follicular gene expression were only present in tonsils. Overall the data presented provide strong evidence that a subset of TFH with a killer phenotype Tfk is present in tonsils (outside the germinal center) and in the peripheral blood.

Comments for the authors:

- The experimental design and sorting strategies to obtain TFH are depicted clearly in supplemental Figure 1.
- It would be helpful to show flow cytometry data on PBMC from the 15 healthy volunteers up front. They are now “hidden” in the Figures 2 and 3 and only results of 10 PBMC are shown in Figure 3.
- Extensive characterization using single-cell RNA sequencing (scRNAseq) and flow cytometry of sorted CD3+CD4+ CD45RA- CXCR5+ cells was done on a few normal PBMC and tonsils as well as

PBMC obtained after vaccination, and tapering cyclosporine in recipients of allogeneic hematopoietic stem cell transplantation. It is therefore difficult to draw general conclusions based on the results of these experiments.

• Suggestions for clarifying the Figures

Figures:

page 17/39, line 337. Introduce the number of cells loaded on 10x genomics.

page 27/39, line 677. Introduce a color bar for Fig 1a.

page 28/39 line 678. Introduce a color bar for Fig 2a.

Supplementary Information:

page 6/19 Unable to read the Supplementary Fig 2, specially Supplementary Fig2.a,b,c,and e.

Page 8/19, line 43. Supplementary Fig 3. Unable to read the letter of the x-axis.

Page 10/19, line 51 Supplementary Fig 4a. Unable to read the letter of the y-axis.

Page 4/19, line 70 Supplementary Fig 6b is cut.

Reviewer #1 (Remarks to the Author):

The authors demonstrated the relationship of CD4+CXCR5+ Tfh cells between secondary lymphoid tissue (tonsil) and peripheral blood using scRNAseq. CD4+CXCR5+ Tfh cells in both shared subclusters, especially cluster 7, which was named Tfk cells here and expressed TIA-1, CXCR3, and GZMB. The scTCR repertoire analysis showed that Tfk cells shared TCRs with other subclusters. Therefore, the authors concluded that Tfk cells represent terminally differentiated cells within Tfh cell subsets, and cTfh cells share phenotypes with their tonsillar equivalents. The results of this study are of high interest to immunologists and clinicians because they could have applications in vaccination. However, unfortunately, the paper seems to lack data to support the authors' conclusions, as one gets the impression that data analysis based on scRNA analysis and the evidence of previous reports has not been fully integrated and adequately described.

Specific comments

(1) In scRNAseq data integration, it seems that cluster 0 needs sub-clustering. In Fig. 1d, the feature map of CCR7, TOX, MYC, S1PR1, and IL-21 shows heterogeneous expression within cluster 0. The authors probably need to identify the optimal FindCluster resolution or explore alternative clustering packages (<https://github.com/lanevskiAleksandr/sc-type>).

Thank you very much for the advice, this is an interesting single cell type identification software / pipeline. It will be very helpful in our future study.

Regarding the cell subclusters, we agree that the non-differentiated, assumed to be progenitor T_{FH} cells (cluster 0) may be further classified into minor groups, however, our finding in this research is that the T_{FH} cells which display cytotoxic properties, in particular cluster 7, and how much the cT_{FH} cells resemble the bona fide T_{FH} cells (cluster 5, 9-11). Cluster 0 is not essentially related to this topic.

Figure R1. Feature plot illustrating gene expression level in UMAP. *CCR7*, *TOX*, *MYC*, *S1PR1*, *IL21* show different expression levels in different cell clusters, however, the difference within cluster 0 is limited.

Nevertheless, to be more careful, we plotted – as you suggested – the gene expression of *CCR7*, *TOX*, *MYC*, *S1PR1*, *IL21* in the combined feature plot (Fig. R1). We think these gene markers display limited heterogeneous expression inside the cluster 0 compared to others, thus we cannot classify different new cell types from it.

To be sure, we further committed a sub-group analysis within cluster 0. As we can notice, c0 may be further divided into 4 subgroups when we exclude all the other cell types (Fig. R2). The heat map suggests there is no obvious difference between the two largest subgroups (subgroup of 0 and 1). The subgroup can be characterized by a slightly stronger gene expression of *ENO3* and *AL138963.4*. The second gene could be interesting, though it has no meaning gene symbol yet, it possibly encodes a translationally controlled tumor protein isoform (TPT1 isoform). The subgroup 3 is relatively different according to its expression profile, however, the population size is rather tiny in the original cluster 0. Therefore, in this study we would keep c0 as one big cluster.

Figure R2. Further analysis using Seurat on c0 (proTFH). Left panel lists the top gene markers for each subclusters, right top panel shows that the original cluster 0 might be divided into four subclusters, the right bottom panel displays the sample distribution (Impf1-4 stand for Vac1-4 in the manuscript).

(2) Both Fig. 1d and Fig. 2a should include annotations for expression levels (indicated by a yellow-to-red heatmap).

Thank you very much for pointing out this problem. We have regenerated the figure scales and merged it into the Fig. 1d and Fig. 2a. All the feature plot panels were visualized using the same cutoff (max cutoff 3) and same scale (Fig. R3).

Figure R3. Scale bar used in the analysis.

We added the scale now in the manuscript (corrected Fig. 1d and Fig. 2a).

(3) In Fig. 2c, the detection levels of *GZMB* vary between PBMC and Tonsil. These FACS plots suggest unequal detection. These plots should be reconsidered.

We thank the reviewer for bringing up this issue. Indeed, GZMB (and to a much lesser extent also TIA-1) staining in tonsils and PBMC look different. However, whereas PBMC were stained immediately after blood drawing, tonsil samples were stained after thawing and incubation overnight of previously cryopreserved material. Thus, constitution of the cellular material differed substantially between the two sample types. To attend to these differences more stringently, we now re-evaluated all gating in Fig. 2b and Fig. 2c and in all samples set the gates for TIA-1 and GZMB in $CD4^+CD45RA^+CXCR5^+$ cells according to the respective expression levels in classical, cytotoxic $CD8^+$ T cells (as exemplarily shown in corrected Fig. 2b, middle panels). This led to minor proportional changes in Fig. 2c but did not change the overall results and conclusions. For additional information, we provide here the original plots for the PBMC and tonsil samples shown in Fig 2c together with the respective plots of $CD3^+CD8^+$ T cells (Fig. R4). We hope, we thereby adequately responded to this request.

Figure R4. TIA-1 and GZMB in $CD4^+CD45RA^+CXCR5^+$ cells according to the respective expression levels in classical cytotoxic $CD8^+$ T cells.

(4) Fig. 4 and Supplementary Fig. 4 depict the relationships of $CD4^+CXCR5^+$ Tfh cells between the tonsil and peripheral blood. In these results, is it possible to align data from the same patient treated with tonsillectomy? It might be better to integrate the Ton1/2/3 and Vac1/2a data without the inclusion of pathological allo-HSCT data.

Figure R5. Aligned sample data of Tonsil group and Vac2 group after batch-effect correction treatment. The left UMAP illustrates the cells from different sample groups, the right UMAP shows they are from different cell cluster/types. Here we can see that the Seurat CCA batch correction method can satisfy the requirement of aligning cells from different groups into a same cell type/cluster for our downstream analysis.

Thank you very much for the suggestion. Unfortunately, we did not have samples of before-tonsillectomy and after-surgery of a same patient. However, in this analysis, we conducted a batch-correction method to align the cells from different sample, so that we could compare them and study the difference between the tonsillectomy-treatment group and PBMC group without treatment. Here we show a separate visualization by integrating data of ton1/2/3 and only Vac2a/b to reveal more details (Fig. R5). The UMAP illustrates that the cells from Ton groups and Vac/PBMC groups can be aligned well, so that we could carry out the analysis.

(5) The authors suggest that Tfk cells are terminally differentiated from classical Tfh cells in Ton2 and CsA1a data. However, RNA velocity in Fig. 7a suggests that cluster 7 is isolated. The authors might consider another trajectory analysis (e.g., Monocle) to improve the isolation of cluster 7.

Indeed, we had performed a trajectory analysis using Monocle 3 (Fig. R6). However, the pseudotime analysis did not give us a clear conclusion. To avoid the interference from other patients' cells, the analysis was carried out on only the cells of Ton2 sample. Cells were re-clustered to acquire a reasonable position in UMAP, where the cytotoxic T_{FH} cells are now located in the right side (Fig. R6), namely cluster 7 and 12. We can see the cytotoxic clusters 7, 12 are very distinct to other cell clusters/types, the left gene expression panels (below, left) also prove that the distances between cluster 7/12 (yellow) and other cell clusters (orange, purple) are huge. This result further proved the conclusion from our RNA velocity analysis.

The reason we did not show this figure in the manuscript is that the pseudotime analysis did not give us any further clear information. In this case, the RNA velocity analysis performs relatively better for studying differentiation purpose.

Figure R6. Pseudotime analysis using monocle3. Left panel are the expression profile of top variable genes along the pseudotime. Right panel are pseudotime analysis result (upper) and cluster figure shown in a UMAP (lower).

(6) It's worth noting that Tfk cells might not be entirely novel, as GZMB+ cytotoxic Tfh cells have already been reported (Ref 60, 61). If the authors intend to differentiate Tfk cells from GZMB+ cytotoxic Tfh cells, more validation experiments, functional assays, and correlation studies on Tfk cells and their pathophysiology are needed.

Yes, we hope we did not make the impression of claiming to be the first describers of T_{FH} cells with a cytotoxic phenotype. We discussed cytotoxic CD4⁺ T cells in general and those two references of cytotoxic T_{FH} cells in particular. It could be that we only analyzed in more depth by our scRNAseq involving only CD4⁺CXCR5⁺ T_{FH} cells and thus could reveal a more pronounced cytotoxic phenotype of the same T_{FH} subpopulation as described before. Still, the predominance of GZMB in the two mentioned publications was replaced by most other granzymes (A, H, K) in our study, making GZMB the least dominant family member, discussed already by us. However, we are more careful with the claim of a new cytotoxic T_{FH} cell type in the discussion now (the paragraph with the former Ref 60, 61, now 69, 70; lines 331-336 on pages 16/17).

In case of a continuum of differentiation or a matter of plasticity (an aspect now included in the discussion), it is at least new that it does not take such severe circumstances as group A Streptococcal recurrent pharyngitis (GAS RT) and severely sick COVID-19 patients to develop killer T_{FH} cells. This is novel by itself, nevertheless making studies for their pathophysiology difficult. Interestingly, however, comparing CsA1a vs b side by side indicated that regular activation of T cells, presumably via their TCR and subsequent NFAT activation (no calcineurin and therefore NFAT inhibition by CsA anymore) limited the frequency of T_{FK} cells (new Fig. 4, b-e, lines 230-244 on page 12, and discussion lines 346-8, p. 17).

In addition, we find the T_{FK} cells predominantly outside of the GC, i.e. at the T-B border (objectively measured and quantitated now, new Fig. 6b), although there is just one representative example in Dan *et al.* for a CD4⁺GZMB⁺ (no CXCR5 or BCL-6 staining) T cell in the follicle to compare. Thus, we did not want to emphasize this difference too much.

Functional assay: Without knowing the antigen specificity or the target cells of T_{FK} cells, we had to come up with an unbiased choice, whereas keeping the MHCII restriction for a functional assay. Thus, we took advantage of SEB as described for the activation of Ag-spec T_{FH} cells in the so-called AIM assay (Dan, J. M. *et al.* A Cytokine-Independent Approach To Identify Antigen-Specific Human Germinal Center T Follicular Helper Cells and Rare Antigen-Specific CD4⁺ T Cells in Blood. *J Immunol* **197**, 983-993, doi:10.4049/jimmunol.1600318 (2016). In line, we added SEB to cryopreserved and thawed tonsillar cells and measured – the significantly enhanced – CD107a expression on T_{FK} cells after 4 days (new Fig. 4a, lines 222-8, p.11).

(7) In the methods section, it would be beneficial to include the GitHub URL for the analysis package.

Thank you very much for the instruction, we have added these GitHub URLs (below) into the current version of manuscript, methods, lines 401 - 436.

GitHub links involved this manuscript:

Seurat: <https://github.com/satijalab/seurat>

Ggplot2: <https://github.com/tidyverse/ggplot2>

Velocity: <https://github.com/velocity-team/velocity.py>

Packcircles: <https://github.com/mbedward/packcircles>

SingleR: <https://github.com/LTLA/SingleR>

ScRepertoire: <https://github.com/ncborcherding/scRepertoire>

ScVelo: <https://github.com/theislab/scvelo>

(8) The authors should accurately reference their sources (e.g., Ref 52, Ref 60).

Sorry, I am not quite sure, what you mean. We wrote, right now starting line 254

Besides, tonsillar CXCR5⁺ Treg cells exposed elevated transcript levels for T_{FR}-typical **IL-1R2**, Treg/T_{FR}-essential CTLA4 and CD25 (*IL2RA*)^{18, 19, 51, 52}

The reference #52 is

52. Ritvo P. G., *et al.* Tfr cells lack IL-2Ralpha but **express decoy IL-1R2** and IL-1Ra and suppress the IL-1-dependent activation of Tfh cells. *Sci Immunol* **2**, (2017).

In the #52 publication they additionally find a lack of CD25, but the given citation #51 documents the gradual loss on T_{FR} cells while they migrate deeper into the GC, i.e. not excluding a CD25 expression *per se*.

51. Wing J. B., *et al.* A distinct subpopulation of CD25(-) T-follicular regulatory cells localizes in the germinal centers. *Proc Natl Acad Sci U S A* **114**, E6400-E6409 (2017).

#60 appeared in this context, currently starting line 326:

Cytotoxic T_{FH} subpopulations with perforin and granzyme B expression were detected in children with group A Streptococcal recurrent pharyngitis (GAS RT) and in severely sick COVID-19 patients among virus-specific CD4⁺ T cells^{60, 61}.

60. Dan J. M., *et al.* Recurrent group A Streptococcus tonsillitis is an immunosusceptibility disease involving antibody deficiency and aberrant TFH cells. *Sci Transl Med* **11**, (2019).

61. Meckiff B. J., *et al.* Imbalance of Regulatory and Cytotoxic SARS-CoV-2-Reactive CD4(+) T Cells in COVID-19. *Cell* **183**, 1340-1353 e1316 (2020).

By the way, under (6) you yourself refer to the references 60 and 61 as describing GZMB⁺ cytotoxic T_{FH} cells already, but probably I just do not understand “accurately reference their sources”.

(9) Some sentences lack clear subjects or have none. Please provide the necessary clarifications.

Sorry about that! We carefully edited the manuscript again and hope to have avoided ambiguities now.

Reviewer #2 (Remarks to the Author):

Summary of research:

The authors study human peripheral blood (PB) and tonsil CD3⁺ CD4⁺ CD45RA⁻ CXCR5⁺ TFH cells by integrative 5' scRNA-seq analysis. They build a dataset of single-cell gene expression, TCR-seq, and CITE-seq of a handful of surface markers for a total of approximately 70,000 cells sorted from 11 distinct tonsil (n=3) or PBMC (n=8) samples. PBMC samples include blood drawn from healthy donors (n=2) before and 7 days after booster vaccine injection, or from allo-HSCT transplant recipients (n=2) before and after cyclosporin A tapering. The authors use a classic single-cell data analysis workflow based on the Seurat analysis tool for the gene expression data, and the scRepertoire analysis tool for the TCR-seq data. After batch integration, the authors find 12 gene expression-based clusters of tonsil and PB TFH cells, which they annotate based on marker genes identity and supervised analysis of key markers of previously known TFH subsets. Most of the clusters are composed of cells from tonsils and PB; three clusters (c5, c10, c11; annotated as GC-II TFH, GC-I TFH, and stress TFH, respectively) are composed only of tonsillar cells. The authors also identify that in clusters shared between tonsil and PB cells, cells from tonsils express higher levels of canonical TFH marker genes, whereas cells from PB express more SELPLG, MYC, CXCR3. One of the TFH clusters (c7, annotated as killer TFH or TFK cells) expresses markers associated with cytotoxic T cells (NKG7, CCL5, various granzymes) and is found equally in tonsils and blood. The authors focus on that TFK subset to confirm its surface phenotype and tissue localization based on the expression of the NKG7/TIA-1 marker: TFK cells express PD1 and CXCR3, and locate outside of GC. Analysis of single-cell TCR-seq reveal that TFK cells are oligoclonal, suggesting clonal expansions in response to some antigen. Analysis of shared TCR clonotypes between TFH clusters within samples reveals that TFK cells share a common origin with other TFH subsets.

Assessment

The study addresses an important question: whether circulating PB TFH cells are good proxies for analyzing TFH responses occurring in secondary lymphoid organs, here tonsils. The results of the study are interesting, based on classic analysis pipelines for that type of single-cell gene expression and TCR-seq data. Thus the results of the single-cell analysis, although descriptive by nature, are robust and of significant interest. The focus on TFK cells is interesting because that subset has been reported occasionally but rarely studied in depth.

Overall, because this is the most original finding, it would be important strengthen the manuscript by providing:

(i) a quantitative analysis of IF images for robust unbiased TFK localization relative to germinal centers and primary B cell follicles,

To address this request, we performed image analysis with Fiji, explained in detail in the method part, line 460-8, p. 22/23: "The image analysis was performed with Fiji (*Fiji is just ImageJ*) for which a macro was designed to incorporate both deconvolution and segmentation

for the images acquired with the confocal microscope. In a next step, cell counting for B cells, CD4⁺ T cells, and BCL-6⁺ cell as well as for TIA⁺ cells was achieved using a pipeline created in CellProfiler, before a second Fiji macro showed the overlap between the segmented images to indicate where T_{FK} cells may be located. To confirm the overlap of the three markers CD4, BCL-6 and TIA-1 in one single cell, intensity profiles along the cell width for the likely cell candidates were created by a third Fiji Macro. For the intensity profiles to confirm the cell candidate as a T_{FK} cell, CD4 needed to be expressed at the cell membrane and overlap with TIA while BCL-6 needed to be expressed within the nucleus.”

We show examples for the intensity profiles in the new Supplementary Fig. 9 and the quantitation in new Fig. 6b.

(ii) more insight into the cytolytic function of TFK cells in *ex vivo* assays

Without knowing the antigen specificity or the target cells of T_{FK} cells, we had to come up with an unbiased choice, whereas keeping the MHCII restriction. Thus, we took advantage of SEB as described for the activation of Ag-spec T_{FH} cells in the so-called AIM assay (Dan, J. M. *et al.* A Cytokine-Independent Approach To Identify Antigen-Specific Human Germinal Center T Follicular Helper Cells and Rare Antigen-Specific CD4⁺ T Cells in Blood. *J Immunol* **197**, 983-993, doi:10.4049/jimmunol.1600318 (2016). In line, we added SEB to cryopreserved and thawed tonsillar cells and measured – the significantly enhanced – CD107a expression on T_{FK} cells after 4 days (new Fig. 4a, lines 222-8, p.11).

(iii) a deeper analysis of the impact of vaccine boost or cyclosporin A tapering on the numbers and gene expression profile of those cells.

We appreciate the suggestion. We have the cell number changes of each cell type in different samples under vaccine boost or cyclosporin A treatment. The result is shown in the bar plot (Fig. 1b and specifically in the new Fig. 4b). The TCR repertoire analysis result shows further differences upon the treatment, contained in the manuscript, now Fig. 7, b-d.

To mine the data for the specific transcriptomic differences caused by vaccination (not overall significant) and CsA tapering (relative loss of T_{FK} cells), we add direct comparisons of two individuals in the new Fig. 4, b-e). In line with less dominant T_{FK}-specific TCR clones, T_{FK}-specific transcripts encoding cytotoxicity were less abundant after the release of CN inhibition. Instead, RNA expression indicated T_{FH} cells being rather naïve, T_{CM} and in G0/G1. New paragraph lines 230-44, p. 11/12 and in the discussion lines 346-8, p. 17.

I also list more specific comments that may be used to enhance the manuscript.

Specific comments:

1. L53 (abstract): the last sentence of the abstract too broad in my opinion when mentioning “counterparts in tissues”. The authors should be more specific: “tonsils”. Thanks for pointing this out, we have corrected it in this version.

2. L68: “High PD-1 expression prevents GC-TFH cell proliferation allowing competition between GC-B cells”: missing reference. This concept is not mentioned in refs 6 or 7. Yes, indeed, this was studied only in detail in the newly added reference (Shi, J. *et al.* PD-1 Controls Follicular T Helper Cell Positioning and Function. *Immunity* **49**, 264-274 e264, doi:10.1016/j.immuni.2018.06.012 (2018).) To be clear, we elaborated a little more on this in the introduction, lines 68-73, p. 4.

3. L71-83 (introduction): the authors may also mention T peripheral helper cells (TPH)

that have a CD4⁺ CXCR5⁻ PD1⁺ phenotype in blood, have B-helper functions, and have been implicated in a number of autoimmune diseases in humans.

We added this description in this manuscript, lines 87-9, p.5.

4. L89-90 (introduction): TFR cells may also derive from TFH cells (iTFR, as reported by Le Coz et al. *Science Immunology* 2023).

Thanks, we have added this notion in the current version, line 97, p. 5, citing Le Coz as well as Aloulou... Linterman (2016).

5. L112-113: the authors should rephrase the statement about the advantage that studying PB before and after CsA tapering brings. As such, this is not understandable.

Sorry, we rephrased it, lines 1119-22, p. 6/7, and added a respective citation.

6. L114: define TEMRA.

The abbreviation has been explained in the current version, lines 123-4, p. 7.

7. L115 (results): the authors should mention in the results section that they performed batch integration as part of their analytic workflow.

Yes, you are right. This has been added into the current manuscript. The CCA batch correction was performed, the alignment UMAP result shows that the batch-effect removal method is effective, lines 125-7, p. 7.

8. L136: it seems that most MYC-expressing cells are not actively proliferating in the dataset.

You are right, as outlined before, best fitting cluster with proliferation is c2. To be clear, we extended the sentence mentioning MYC: "...with an enhanced likelihood of entry into cell cycle (MYC), but no indication of active proliferation." Lines 141-2, p. 7

9. Cluster annotation: there is little detail about how the clusters were named or how the selected genes confirm cluster identification (eg. how was cluster 5 assigned a GC-Tfh label?). It would be interesting to investigate differentially expressed genes between GC-I and GC-II clusters.

In addition to the experimental observation (figures in manuscript), all the cell clusters are further validated and annotated according to the transcriptome profile, they are then examined using PanglaoDB.

And sure, the different clusters containing *bona fide* GC-T_{FH} cells are not only obvious, but very interesting, especially as they are not even neighbors in the UMAP. Nevertheless, they share some TCR clones.

We tried to explain our naming when we first describe the clusters in the result section, providing citations for each gene in the context of follicular T-cell differentiation. Thus, investigations for the differentially expressed genes have been performed by others (and us for NFATc1). We believe that any additional experiments in the context of GC-I vs GC-II – as we had seven, now eight figures and even more supplemental figures – would go beyond the scope of this manuscript comparing tonsil and periphery, and then concentrating on the novel aspect of cytotoxic T_{FH} cells.

To name GC-I and GC-II in this order seemed appropriate as GC-II has even more canonical T_{FH} genes upregulated (described in line 156-172, p. 8/9).

10. L142: precise "type 1 interferon response".

You are right and we specified it, line 178, p. 9.

11. L144: define “a classical TFH gene expression pattern.

This comes back to your request #9. We exchanged ‘classical’ to ‘canonical’, an expression you use below (#12), while keeping the cited review article by Shane Crotty and adding the two papers mentioned by you. The explanation is then given in the paragraph by listing the respective genes.

12. L212: “...T cells derived from tonsils were more TFH-like than their cTFH counterparts”. The authors are analyzing only TFH-phenotype cells, so the “TFH-likedness” concept is a bit blurry and not properly defined here. It should be properly defined that those genes are part of the canonical TFH marker genes, as observed in other studies of human TFH cells (King et al. Science Immunology 2021, Le Coz et al. Science Immunology 2023).

We try to be less blurry, rephrased the sentence and cited Le Coz as well as Crotty, 2019. The mentioned King et al. – as interesting this paper is in this context – rather defines B cells. “In comparison to their cT_{FH} counterparts in the same cluster, CD3⁺CD4⁺CD45RA⁻CXCR5⁺ T cells derived from tonsils mostly displayed a canonical T_{FH}-transcriptome^{1,25} as exemplified by transcripts for TOX, TOX2, BATF, CXCR5, PD-1, CD200, ICOS, SAP (*SH2D1A*), FOS, JUNB and NURR1 (*NR4A2*), which was highly significant in most clusters.” Lines 250-4, p. 13.

13. L226-227: there is a discrepancy between the statement “but never within a GC” in the text, and the title of Supplementary Figure 6 (“only some TFK cells locate at/within GC”). A broad quantitative analysis of TFK-phenotype cells in IF images from several tonsil sections of different human donors is required to conclude.

You are right, there was a discrepancy in our formulation. We corrected the statement and, importantly, provide quantitation:

“Consistent with *SELPLG* and *CXCR3* expression observed in all cells of c7, we could unambiguously detect single T_{FK} cells localized close to the edge or outside GCs. In a more objective way, we quantitated T_{FK} cells within GCs in comparison to their occurrence at the T-B border (new Supplementary Fig. 9). While some images showed no T_{FK} cells at all, they were overall significantly higher at the T-B border than in the follicle itself (new Fig. 6b).” Lines 264-9, p. 13.

14. L231: what is the quantitative measure of the “highest degree of clonality”? The term “clonality” is ambiguous as it could refer to clonal size, clonal diversity, clonal relatedness, etc.

Yes, it stands for the highest degree of clonal size. We have specified it in the manuscript, line 273, p. 14.

15. L247: “c0 proved to be the proTfh cells for the classical c5, c9 and c10”. TCR overlap does not provide information on the directionality of the relationships between clusters.

We agree with the reviewer, this might be misleading. c0 is proven to be proT_{FH} based on the transcriptome analysis. We rephrased: “Nevertheless, c0 characterized as proT_{FH} cells and sharing TCRs with the classical c5, c9 and c10 within the tonsils, suggest a development from c0 to *bona fide* T_{FH} cells (Fig. 8b).” Lines 289-91, p. 14.

16. L266: “proTFH expressed the entire set of classical TFH genes, questioning whether Tfh differentiation occurs in parallel at both sites”. The phrasing is quite confusing. According to several publications (eg. He et al Immunity 2013 ref 55), precursor TFH

emerge in the T zone and some of them reach the efferent lymph and bloodstream, thus forming an early memory pool in the bloodstream, while the others reach the T:B border for full differentiation. Does this dataset question these results or provide more consistent ones?

No, we do not question these results. We had mentioned already in the introduction that “genetic studies suggest that CXCR5⁺CD4⁺ cT_{FH} cells are predominantly generated from cells committed to the T_{FH} lineage, but not from GC-T_{FH} cells”. There is hardly an argument against genetic studies. Accordingly, we had discussed and cited He et al, reference #55 (now 64) in the mentioned context in the discussion.

It was meant as a prelude sentence based on the different phenotypes of proT_{FH} cells from tonsils and blood we found. Building on this, we wanted to discuss the further differentiation, a word which is now included in the respective sentence in the discussion, because there seemed to be several subtypes (cluster) of T_{FH} cells before they reach the final GC-T_{FH} point of differentiation. All the intermediate phenotypes were detected at both sites, interestingly with site-specific characteristics. So, is it a variable release of different subtypes from SLOs – except of course of the GC-T_{FH} cells – or do those subtypes differentiate (also) in periphery from proT_{FH} cells, but have the capacity to home to SLOs, which allows BLC-6 upregulation and thereby more T_{FH}-typical genes? One possibility to address this question in the future would be to sequence (scRNA and TCR) matched samples from tonsils and blood of the same donor. For now, we hope that the addition of the word “further” clarifies our intention in the discussion (line 310, p. 15).

17. L298: in order to see whether the TFK clones may be specific for EBV or CMV antigens, the authors could try mapping their TCR CDR3 sequences to public repositories of antigen-specific TCR sequences (www.iedb.org).

The top expanded clonotype (**CASSLEAASSYNEQFF**) was identified to be SARS-COV2 (unlikely because the sample was donated before the pandemic) by TCRMatch software, whereas VDJDB prediction is CMV. Unfortunately, as the result differs from each other, we could not reach a concrete conclusion in the manuscript.

In general, there are less data available for MHCII-restricted TCRs, making our sequences even more difficult to predict a specificity.

18. Fig1d: color legend is missing.

We added the legend in the revised version.

19. Fig2a: color legend is missing.

The color legend was added in the current version.

20. Fig3b: color legend is missing.

We have regenerated the figure using 4 columns, now it includes a color panel legend.

21. Fig3c: Why are the authors not comparing between CXCR5⁺TIA⁺ and CXCR5⁻TIA⁻?

According to this comment, we include detailed comparisons between all CD45RA⁺/⁻ CXCR5⁺/⁻ TIA⁺/⁻ cells (Fig. 3d and new Supplementary Fig. 4), lines 218-20, p. 11.

Are CXCR5 levels continuous or bimodal in CD4⁺TIA1⁺ cells?

An interesting question, indeed. To demonstrate bimodal CXCR5 expression level in CD4⁺CD45RA⁻ T cells, we plotted TIA against CXCR5, where bimodality could be detected among the CXCR5⁺ population (Fig. R7). If you do not insist, we are not including it in the already crowded manuscript.

Figure R7: Cells were isolated from PB by Ficoll, gated for $CD3^+ CD4^+ CD45RA^-$ and CXCR5 was plotted against TIA.

22. Fig6a: please include C1 instead of 1 etc. Otherwise it looks like a ratio. What does the 2nd number correspond to?

We include the 'c' in the figure (now Fig. 7a) as well as the corresponding supplementary figure to indicate clusters. The second number was put in brackets to not confuse it with a ratio. It gives the number of cells appearing in the individual clusters having functional TCRs. The latter is mentioned in the respective figure legend and explained in the method paragraph 'Single-cell TCR repertoire profiling' starting line 415, p. 20.

23. Fig6d: define Morisita index.

Sorry, we used the abbreviation instead of the full name. We apologize for it. Now Fig. 7d

Morisita index here stands for an overlap coefficient, the full name is Morisita's overlap index. This statistical measure of dispersion is used for comparing overlap among different samples.

Ref: Morisita, M. (1959). "Measuring of the dispersion and analysis of distribution patterns". Memoires of the Faculty of Science, Kyushu University, Series E. Biology. 2: 215–235 [no longer in PubMed and thus not citable]

Now included in the M&M section: The morisita's overlap index measures how many times it is more likely to randomly select two sampled points from the same quadrat (the dataset is covered by a regular grid of changing size) then it would be in the case of a random distribution generated from a Poisson process. Duplicate objects are merged with their counts summed up. Lines 426-30, p. 21.

24. Fig6e: colors in bubble plots and legend do not match (killer TFH notably).

Thanks for pointing this out, we adjusted the color in now Fig. 7e.

25. Fig7c: those series of Venn diagrams are very hard to read. The authors could use a

heatmap display for clonotype overlap, as was used in King et al. Science Immunology 2021.

We agree that such bigger data sets are often hard to read. We prefer to keep the Venn diagrams, but enlarged the font size and importantly, deleted all unnecessary 'zeros' and 'zero percentage'. Additionally, TCR clones shared by c7 and any other cluster are indicated in red. We hope that the implication is easier to grasp now (changed to Fig. 8c).

26. Supplementary Fig3: the authors should include horizontal bars color coded with sample origin (PB or tonsil, sample ID).

Thank you very much for the advice, it is a nice idea. However, the current version of Seurat DoHeatmap does not offer this multiple bar function to plot sample sources (orig.ident) into a second bar below the cell cluster bar. We apologize for this; we are not able to plot it with the current software version. (meanwhile Fig. S5)

Reviewer #3 and #4 (Remarks to the Author):

Oligoclonal CD4+CXCR5+ T cells with a cytotoxic phenotype appear in tonsils and blood.

Summary:

The authors examined subsets of CD4+CXCR5+ T follicular helper cells (TFH) in the peripheral blood and tonsils to determine whether peripheral blood can be used as a mirror of the TFH cell subsets in lymphoid tissues. Extensive analysis included examination of transcriptomes, TCR repertoire, cell surface expression using single-cell RNA sequencing (scRNAseq) and flow cytometry of sorted CD3+CD4+ CD45RA-CXCR5+ cells. Immunohistochemistry was also performed. The peripheral blood of 2 healthy individuals was examined before and after vaccination. Tonsils were obtained from 11 individuals with moderate tonsillitis. Three tonsils were extensively examined with scRNAseq, the other 8 were used for Flow Cytometry and histology. In addition peripheral blood was examined from 2 individuals after allogeneic hematopoietic stem cell transplantation before and after the decreasing cyclosporin. Blood from 15 healthy donors was used for flow cytometry alone. The results of these studies show that similar subsets of CD4+CXCR5+ cells are present in the peripheral blood and tonsils including an additional subset with a cytotoxic phenotype (Tfk). Three subsets of CD4+CXCR5+ cells with T follicular gene expression were only present in tonsils.

Overall the data presented provide strong evidence that a subset of TFH with a killer phenotype Tfk is present in tonsils (outside the germinal center) and in the peripheral blood.

Comments for the authors:

- The experimental design and sorting strategies to obtain TFH are depicted clearly in supplemental Figure 1.

Thank you.

- It would be helpful to show flow cytometry data on PBMC from the 15 healthy volunteers up front. They are now "hidden" in the Figures 2 and 3 and only results of 10 PBMC are shown in Figure 3.

First of all, we apologize for not counting right, we only analyzed 13 anonymous healthy donors, merely together with the vaccines, we had 15 healthy donors. We specified this in the M&M section now and corrected the table S1.

Regarding your wish to show those PBMC data up front, we humbly contradict. A representative example of PBMC in comparison to a tonsil is given in Suppl. Fig. 1. The actual experiments then begin with an unbiased single cell sequencing, thereby avoided flow cytometric analysis by known markers. Important findings are only then verified in Fig. 2 und Fig. 3, but still rather early in the manuscript.

- Extensive characterization using single-cell RNA sequencing (scRNAseq) and flow

cytometry of sorted CD3+CD4+ CD45RA- CXCR5+ cells was done on a few normal PBMC and tonsils as well as PBMC obtained after vaccination, and tapering cyclosporine in recipients of allogeneic hematopoietic stem cell transplantation. It is therefore difficult to draw general conclusions based on the results of these experiments.

It might be a small cohort of healthy individuals, but a general view on T_{FH} cells in tonsils vs PBMCs could still be described. To mine the data for the specific transcriptomic differences caused by vaccination (not overall significant) and CsA tapering (relative loss of T_{FK} cells), we add direct comparisons of two individuals in the new Fig. 4, b-e). In line with less dominant T_{FK}-specific TCR clones, T_{FK}-specific transcripts encoding cytotoxicity were less abundant after the release of CN inhibition. Instead, RNA expression indicated T_{FH} cells being rather naïve, T_{CM} and in G0/G1. New paragraph lines 230-44, p. 11/12 and in the discussion lines 346-8, p. 17.

- **Suggestions for clarifying the Figures**

Figures:

page 17/39, line 337. Introduce the number of cells loaded on 10x genomics.

We now include in the M&M section: 'Freshly sorted human T_{FH} cells (between 11,000 and 20,000 per sample) were loaded ... ', line 390, p. 19. Further, the sequenced cell numbers appear in the result part. In total 72,608 cells were identified in this scRNAseq analysis, line 137, p.7.

page 27/39, line 677. Introduce a color bar for Freshly sorted human T_{FH} cells (between 11,000 and 20,000 per sample) were loaded Fig 1a.

Fig. 1a shows the cell clusters marked in different color, the cells are plotted in the UMAP. Here we categories and color codes are shown in the figure.

page 28/39 line 678. Introduce a color bar for Fig 2a.

We have added a scale bar into Fig. 2a.

- **Supplementary Information:**

page 6/19 Unable to read the Supplementary Fig 2, specially Supplementary Fig2.a,b,c,and e.

Thanks for pointing this out. In the current version, we regenerated the suppl. Fig. 2a, piechart in a vector format. The default font size was much increased, too. We further transferred S2e into a separate figure – now Supplementary Fig. 3 – which gave us the opportunity to enlarge all labels of the former Supplementary Fig. 2.

Page 8/19, line 43. Supplementary Fig 3. Unable to read the letter of the x-axis.

Here it is readable when enlarged to 100 % on the screen, but we have a high-resolution figure for the final production to clear sufficiently, now Supplementary Fig. 5.

Page 10/19, line 51 Supplementary Fig 4a. Unable to read the letter of the y-axis.

We newly provided high resolution figures, which should be sufficient to read when enlarged to 100 %, now Supplementary Fig. 6a.

Page 4/19, line 70 Supplementary Fig 6b is cut.

Thanks for noticing, it's fixed now (now Supplementary Fig. 8b).

Reviewers' comments:

Reviewer #2 (Remarks to the Author):

The authors have addressed all my specific comments carefully and included new analyses related to the TFK subset that strengthen the manuscript.

Reviewer #3 (Remarks to the Author):

Reviewer 1

#4. Figure 4 and supplemental Figure 4

Regarding the sentence: "Samples from same individuals PBMC vs. tonsil are not available".

Comment: Batch correction is not accurate because there are not the same controls that are run in every flow cytometry experiment. This inhibits detecting a possible biological effect.

Reviewers 3,4

All our comments have been answered satisfactory.

Reviewer #4 (Remarks to the Author):

The authors has answered all the questions. Just an aesthetic, Figure 1 D and similar: cluster 3 and 0 are too closed , they looks as 30.

Reviewer #3 (Remarks to the Author):

First of all, many thanks for taking the extra effort to assess our former rebuttal to the comments made by reviewer #1.

Reviewer 1

#4. Figure 4 and supplemental Figure 4

Regarding the sentence: "Samples from same individuals PBMC vs. tonsil are not available".

Comment: Batch correction is not accurate because there are not the same controls that are run in every flow cytometry experiment. This inhibits detecting a possible biological effect.

Answer:

Many thanks for pointing this out. You are right, without control groups, people are not able to remove the batch-effect perfectly, in particular for bulkRNAseq and other analyses based on cytometry experiments.

In this study, unfortunately we had only scRNAseq data from different samples (tonsil, Vac and CsA), so that we applied the Seurat-Canonical Correlation Analysis (CCA) method to correct the batch-effect as it is currently one of the most popular methods in scRNAseq data analysis. Many other research groups encounter this same problem/restriction as we and normally use CCA/MNN methods or other algorithm to minimize the batch-effect derived from experimental design. Although the batch effect remained still to be observed, it had been reduced sufficiently to allow further downstream analysis. Let us show you the 'before' and 'after'.

2nd revision Figure 1. Seurat clustering result shown in UMAP.

In **2nd revision Figure 1**, the clustering result before we minimized the batch-effect is demonstrated. Left panel indicates there are in total 16 cell clusters (types), and the right panel shows these cells are from different isolation/libraries. Here we can clearly see that the batch-effect is huge. All the tonsillar cells are located in the left island (orange, light green and blue dots in right panel), meanwhile most of CsA cells and Vac cells are also separated from each other. This observation proved that direct clustering without batch-effect correction is hardly convincing.

To overcome the obstacle, we applied the Seurat-CCA method, the math behind is actually “dual PCA”. The anchor genes were determined for data integration, followed by FindNeighbors, RunPCA and FindClusters. The final clustering outcome is demonstrated in **2nd revision Figure 2**. The cell cluster number was reduced to 12 from 16. Although the batch-effect is still detectable, the right panel (Figure 2) suggests that the batch-effect has been greatly reduced compared to before (Figure 1). Next, we investigated the differences and found that they are mostly from the cells extremely enriched in either tonsil or blood.

2nd revision Figure 2. Seurat clustering result after CCA shown in UMAP.

Reviewer #4 (Remarks to the Author):

The authors has answered all the questions. Just an aesthetic, Figure 1 D and similar: cluster 3 and 0 are too closed, they looks as 30.

Answer:

Both Figure 1d and Figure 2a have been corrected, now the cluster 3 and cluster 0 can be distinguished by label. Please, see also below.

Figure 1d

Figure 2a

REVIEWERS' COMMENTS:

Reviewer #3 (Remarks to the Author):

The authors answered all the questions previously and improved Figures 1d and 2a.

In Figures 1d and 2a, the clusters 3 and 0 were depicted too close and looked like a cluster 30. The authors corrected both Figure 1d and Figure 2a so that the labels of cluster 3 and cluster 0 are clearly separated.